# A perceptual scaling approach to eyewitness identification

Sergei Gepshtein [1,2✉], Yurong Wang[1,3], Fangchao He[1,4], Dinh Diep[1] & Thomas D. Albright [1✉]

Eyewitness misidentification accounts for 70% of verified erroneous convictions. To address this alarming phenomenon, research has focused on factors that influence likelihood of correct identification, such as the manner in which a lineup is conducted. Traditional lineups rely on overt eyewitness responses that confound two covert factors: strength of recognition memory and the criterion for deciding what memory strength is sufficient for identification. Here we describe a lineup that permits estimation of memory strength independent of decision criterion. Our procedure employs powerful techniques developed in studies of perception and memory: perceptual scaling and signal detection analysis. Using these tools, we scale memory strengths elicited by lineup faces, and quantify performance of a binary classifier tasked with distinguishing perpetrator from innocent suspect. This approach reveals structure of memory inaccessible using traditional lineups and renders accurate identifications uninfluenced by decision bias. The approach furthermore yields a quantitative index of individual eyewitness performance.

---

[1] Center for the Neurobiology of Vision, Salk Institute for Biological Studies, 10010 North Torrey Pines Road, La Jolla, CA 92037, USA. [2] Center for Spatial Perception and Concrete Experience, School of Cinematic Arts, University of Southern California, 3470 McClintock Avenue, Los Angeles, CA 90089-2211, USA. [3] Division of Biological Sciences, University of California San Diego, La Jolla, CA 92037, USA. [4] Division of Biological Sciences and Department of Bioengineering, University of California San Diego, La Jolla, CA 92037, USA. ✉email: sergei@salk.edu; tom@salk.edu

Eyewitness identification has long played a valuable role in criminal investigations and prosecutions. Despite this value, our society has been confronted in recent years with many rank failures of eyewitness testimony[1]. For example, more than 350 people, many serving lengthy prison sentences, have been exonerated in the United States because their DNA was found to be incompatible with evidence collected from the crime scene. In ~70% of these cases, misidentification by eyewitnesses contributed significantly as evidence for conviction[2].

It is natural to ask what can be done to improve the performance of eyewitnesses, such that they are more likely to identify the culprit and less likely to misidentify an innocent person[3,4]. A major focus of research on this topic has been the manner in which an eyewitness lineup is presented[5–11]. The traditional simultaneous (SIM) lineup is composed of (typically) six facial photographs shown at the same time (Fig. 1a). One of the faces is that of the suspect and the others, known as fillers, are of people known to be innocent. The alternative sequential (SEQ) lineup involves presenting the photographs one at a time (Fig. 1b). In both lineup types, witnesses are asked to identify the perpetrator or to reject the lineup if no face matches the memory from the crime scene.

The performance of eyewitnesses under the SIM and SEQ paradigms reflects recognition memory[12], a form of declarative memory retrieval in which a sensory cue stimulus elicits the trace of a previous encounter with the stimulus. This recognition memory signal is the basis upon which an eyewitness decision is made. The eyewitness also employs a decision criterion: only recognition memory signals that meet this criterion lead to identification. Because this recognition process is covert, the overt response ("that's the culprit") confounds the strength of the recognition memory signal with the decision criterion, leaving the outcome susceptible to unrecognized bias.

Recent laboratory studies of SIM and SEQ lineups have attempted to overcome this problem using expressed confidence as a proxy for the eyewitness decision criterion or by introducing explicit biasing instructions[8–12]. This approach has enabled some important insights into eyewitness memory under different lineup conditions[12–16]. A desirable alternative is to estimate recognition memory signals themselves, which would enable criterion-independent analyses of the relative strengths of these signals. We propose here a lineup procedure that does so. Our procedure employs two powerful experimental techniques that are rooted in the history of scientific study of perception and memory: perceptual scaling and a signal detection method known as receiver operating characteristic (ROC) analysis.

Broadly considered, the goal of perceptual scaling is to map the relationship between a set of physical stimuli and the corresponding responses of an observer's perceptual system. The goal of signal detection analysis in this context is to determine the optimal performance of a perceptual system tasked with classification of a set of stimuli. Used together, these tools enable us to scale the strengths of recognition memory signals elicited by lineup faces, and to quantify the best possible discriminability of those signals. Although these tools have been previously combined in laboratory studies of perceptual discrimination[17], they have not been applied jointly to the eyewitness identification problem.

We utilize this approach to reveal detailed structure of eyewitness recognition memories for a complete set of lineup faces. We then use these memory signals to measure how well an optimal statistical classifier can distinguish the perpetrator from an innocent suspect. Performance of the optimal classifier compares well to traditional lineup procedures, with the added benefits that identification decisions made by the classifier are not subject to unrecognized bias, that it renders a numerical index of performance for individual eyewitnesses, and that lineup filler choices are quantitatively assessed for fairness. For all of these reasons, this approach has enormous potential as a research tool for evaluating effects of other variables on eyewitness performance and as a practical tool for unbiased investigation and prosecution of crimes.

## Results

**Scaling of memory signals elicited by lineup faces**. Our scaling procedure employs Louis Thurstone's Method of Paired Comparisons[18], which is a longstanding behavioral technique designed to scale stimuli along a psychological continuum, such as perceptual similarity to a target stimulus. In essence, the method estimates the central tendency and variance of the perceptual signals elicited by each stimulus, from which it is possible to draw criterion-independent conclusions about perceived similarity. This method has been used in many practical applications, such as product marketing and preference testing[19,20], optometric

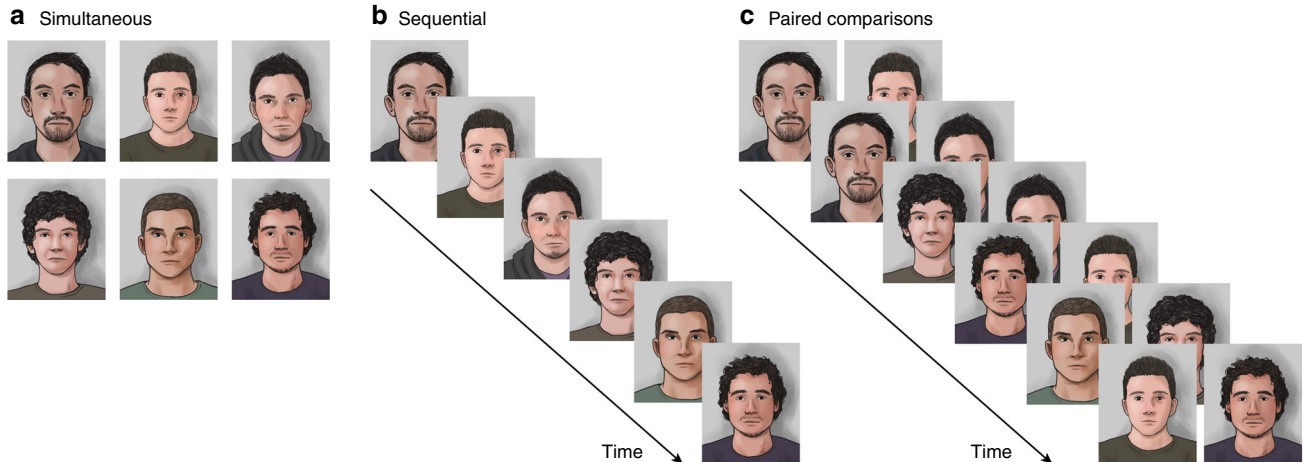

**Fig. 1 Three eyewitness lineups. a** Simultaneous (SIM) lineup, in which six facial photographs are presented at the same time. **b** Sequential (SEQ) lineup, in which six photographs are presented one at a time. In both SIM and SEQ lineups, subjects are asked to identify the perpetrator or to reject the lineup if no face is recognized from the crime. **c** Paired comparison (PAR) lineup, in which facial photographs are presented in pairs. Upon viewing each pair, subjects are required to report (in a two-alternative forced choice task) which of the two faces is more similar to memory of the perpetrator. For the present experiment there were 15 possible pairs; each pair was presented three times in random order (a small subset of face pairs is shown in **c**).

assessment ("which is clearer: lens one or lens two?")[21], and hearing-aid evaluation[22]. In cases where the target exists in memory alone, such as with eyewitness identification, the method estimates the central tendency and variance of recognition memory signals elicited by the cue stimuli.

Features of the paired comparison approach can be shown by our application of the method to the eyewitness identification problem. Subjects were initially presented with a video of a mock crime, followed by a delay and then presentation of a lineup. The paired comparison (PAR) lineup consisted of three presentations of all possible pairings ($N = 15$) of six face images (designated F1–F6), one of which (F1) was the suspect (Fig. 1c). Upon viewing each pair, each subject was required to report which of the two faces was more similar to their memory of the perpetrator. Each report, or vote, is thus a relative judgment, not an absolute identification of the sort rendered in a SIM or SEQ lineup. The complete set of votes was used to compute a voting score, which quantifies the perceived similarity of each face to the perpetrator.

To illustrate perceptual scaling, we present the similarity matrix for one subject in Fig. 2a. Each cell in the matrix corresponds to the indicated pair of faces and contains the sum of votes to all presentations of that pair. For example, the values of two and one in respective cells [3, 2] and [2, 3] indicate that F3 was preferred over F2. The voting score for each face is equal to

the sum of votes across all comparisons with each of the other five faces, i.e., down each column of the matrix (Fig. 2b), normalized by the number of repetitions (Fig. 2c).

**Conditions of lineup composition.** Superior lineups are naturally defined as those that lead to a high probability of correctly identifying the perpetrator (target) and a low probability of incorrectly identifying an innocent suspect[5]. We evaluated our PAR lineup procedure relative to this standard using two distinct lineup conditions that have been routinely employed for this purpose in traditional SIM and SEQ lineups[5,8]. In the target-present (TP) condition the target (F1) was included in the lineup, along with five fillers who matched the physical description of the perpetrator. The target-absent (TA) condition differed from the target-present condition solely in that the target was removed and replaced by another face (F1*) that matched the physical description of the perpetrator. PAR lineups were administered for both TA and TP conditions and the perceptual scaling analyses described below were performed separately for each condition. Results of scaling in TA and TP conditions were subsequently combined using a signal detection procedure described in the section "Signal detection analysis of recognition memory signals" below. This procedure yielded a criterion-independent measure of the probability that an optimal binary classifier can distinguish between the perpetrator and an innocent suspect.

**Target-present lineup condition.** We used our scaling procedure initially to determine the top-ranked face within the set of six TP PAR lineup faces (Fig. 2e), which is a simple analog of the identification choice in traditional TP SIM and SEQ lineups. The frequency distribution of top-ranked faces for the PAR lineup appears in Fig. 3a. If subjects had no discriminative ability, each face would be top-ranked equally often (16.7% of the time). In fact, the target face (F1) was most commonly top-ranked ($n = 20$; 32.26%), significantly exceeding chance performance ($p = 0.016$). By this simple measure, the target face was "identified" in the PAR lineup at a rate comparable to that in the SIM (Fig. 3b) and SEQ (Fig. 3c) procedures, but the number of lineup rejections (see Methods) in the PAR lineup ($n = 5$; 8.06%) was significantly lower than in the traditional lineups (SIM: $n = 16$, 48.48%; SEQ: $n = 24$, 64.86%; $p < 10^{-5}$).

Although generally supportive of the paired comparison approach, this top-rank frequency analysis neglects the richness of data produced by perceptual scaling and reveals nothing about the consistency of recognition memory signals elicited by each face or the degree to which signals vary between faces. In particular, top-ranked status alone does not address the extent to which that face is scaled significantly above all of the other faces in the lineup.

To illustrate these additional features of the PAR data, we asked what the relative rankings were within the set of six TP lineup faces for the entire population of subjects assigned to that condition ($N = 62$). Average voting scores are plotted in Fig. 4a. As mentioned in the caption of Fig. 2, consistency of votes across different presentations of each face is inversely proportional to the variance of recognition memory signals elicited by that face. This variance is represented in Fig. 4a by vertical bars that mark interquartile ranges of the vote distributions for every face. Average voting scores together with voting variance enable us to evaluate the significance of differences between the rankings of each lineup face. Face F1 was by far the top-ranked face ($p = 0.00002$), on average, which reflects the fact that this target face consistently scored highly, even though it was not the top-ranked face for every subject.

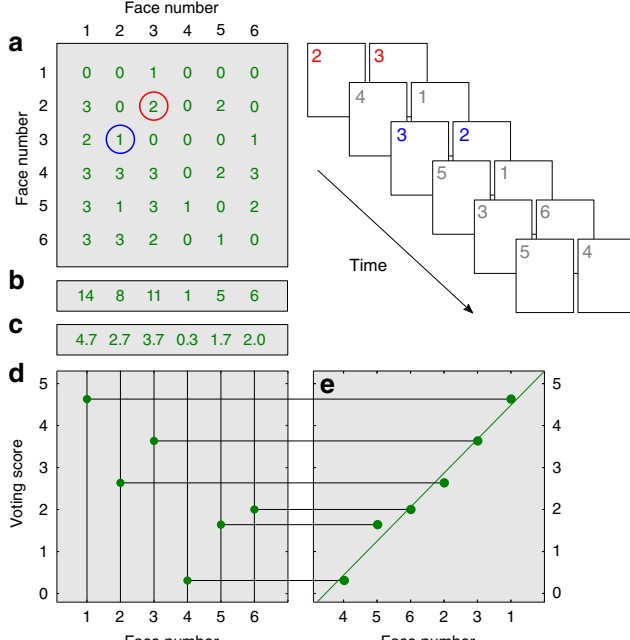

**Fig. 2 Method of paired comparisons applied to eyewitness identification.** Subjects were presented with a sequence of face pairs shown schematically at top right (here using six numerals). The task was to indicate which face in each pair was more similar to the remembered face of the perpetrator. **a** Outcomes of pairwise comparisons were recorded in a voting matrix. The matrix displayed contains results of 45 paired comparisons (15 pairs each presented three times) for one subject, S15. **b** The sum of votes in each column is the cumulative vote for every face. **c** Dividing the cumulative vote of every face by the number of face repetitions in the lineup yields the normalized voting score for every face (**c**). Voting scores range from zero (face was never selected) to five (face was selected every time it was presented). **d, e** Voting scores are plotted as a function of ordinal face number in (**d**), and as a function of face number ranked by the voting score in (**e**). The latter plot is the voting function, whose slope reflects consistency of voting.

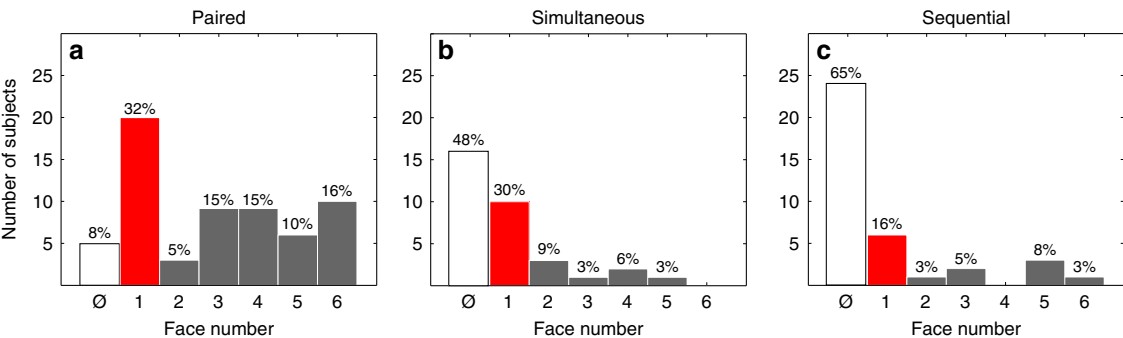

**Fig. 3 Results of three lineup procedures.** Frequency distributions of subjects selecting the six faces used in three target-present lineups: paired comparisons (PAR; $N = 62$ subjects) in (**a**), simultaneous (SIM; $N = 33$ subjects) in (**b**), and sequential (SEQ; $N = 37$ subjects) in (**c**). Symbol ø in the abscissa of every plot represents the outcome in which subjects made no selection (i.e., "rejected" the lineup). In every lineup, more subjects correctly selected the target (F1, red bar) than other faces. Proportion of correct identifications was largest in paired comparison lineup.

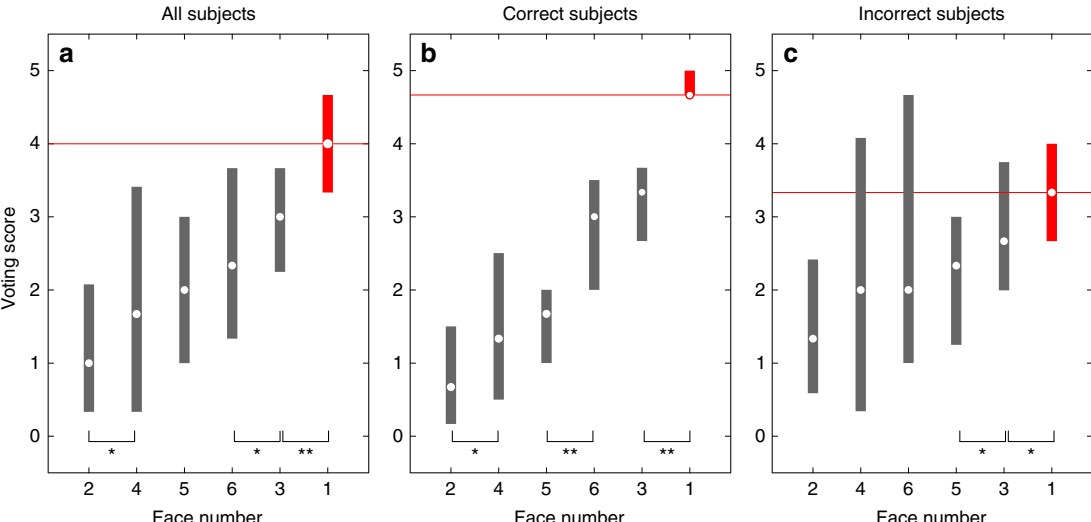

**Fig. 4 Results of perceptual scaling in the target-present condition.** Structure of recognition memory is revealed by voting scores obtained from the paired comparison (PAR) target-present lineup: for all subjects in (**a**) ($N = 62$ subjects, who also appear in Fig. 3a), for subjects who exhibited top-rank voting scores for the target face (F1) in (**b**) ($N = 20$ subjects), and for subjects who exhibited top-rank voting scores for non-target faces in (**c**) ($N = 37$ subjects). (The five subjects, for whom there were voting score ties for top rank, are included in (**a**), but not in (**b**) or (**c**).) Circles represent medians. Vertical bars represent interquartile ranges. Significant differences between the faces in adjacent ranks are indicated by single ($p < 0.05$) or double ($p < 0.01$) asterisks. A nonparametric statistical test was used to evaluate differences, as described in Methods (section Perceptual scaling of lineup faces). This is a one-sided statistical test.

We further divided these data by correctness of subject responses. Voting scores for subjects who preferred target face F1 (correct subjects) are plotted in Fig. 4b and scores for subjects who preferred other faces (incorrect subjects) in Fig. 4c. As expected, the average voting score for the target face greatly exceeded the scores for other faces amongst subjects who performed correctly (Fig. 4b) ($p < 10^{-6}$). Notably, the average score for target face F1 was also highest amongst subjects who performed incorrectly (Fig. 4c) ($p = 0.04$). This counterintuitive finding reflects high consistency of target face judgments among our subjects. (Subjects in the incorrect group often ranked face F1 as their second choice.) More generally, the observed scaling of voting scores reveals informative structure of recognition memory signals not accessible using traditional lineups.

The data shown in Fig. 4 illustrate perceptual scaling of lineup faces averaged across the population of subjects exposed to our target-present PAR lineup. This population analysis is essential for comparison with laboratory studies of eyewitness performance under traditional SIM and SEQ lineup conditions, which are necessarily based on population analysis. Another unique

feature of our PAR approach, however, is that it can be applied to individual subjects. In such cases, the average recognition memory signal for each lineup face is estimated from the corresponding voting score. Variance of each recognition memory signal is estimated from the consistency of votes that underlie each voting score, which is in turn reflected in the slope of the voting function obtained from each subject (see Methods for details). Results of scaling for three individual subjects in the target-present PAR condition are illustrated in Fig. 5. Low voting variance is manifested as high slope (e.g., subject S55) and high voting variance is manifested as low slope (e.g., subject S44).

Yet another feature of perceptual scaling is that it yields a quantitative index of the degree to which the lineup is "fair," i.e., not biased in favor of identification of one or another face. Lineup fairness can significantly impact eyewitness performance[8,23,24], and a number of strategies have been employed to evaluate and improve lineup fairness[7,25,26]. Perceptual scaling of lineup faces reveals the perceptual similarity of those faces, allowing for insight into the degree to which a lineup is fair or biased. Specifically, the means and variances of voting scores can be used

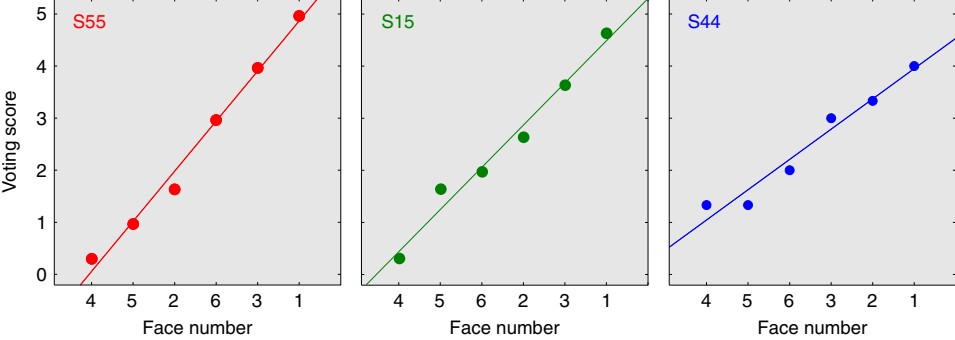

**Fig. 5 Individual voting functions.** Examples of voting functions for individual subjects. Receiver operating characteristics for the same subjects are displayed in Fig. 8, matched by color.

to quantify the extent to which lineup faces resemble one another. This face similarity information can be taken in to account as a factor that influences eyewitness performance. Even as we attempted to construct a fair lineup, perceptual scaling revealed that our lineup fillers resemble the target to varying degrees (Fig. 4), which is evidence of an "unfair" lineup. We draw upon this measure of lineup fairness below as we evaluate performance using a signal detection procedure.

**Target-absent lineup condition**. The target-absent condition was composed of a PAR lineup identical to the target-present condition, with the exception that the target face was removed and replaced by another face (F1\*) that matched the physical description of the perpetrator. The purpose of this manipulation was to determine, in conjunction with the TP condition, the relative probabilities of selecting the target vs. a non-target face. As for the TP condition, we asked what the relative rankings were within the set of six TA lineup faces for the entire population of subjects assigned to that condition ($N = 70$). Average voting scores are plotted in Fig. 6. Variance is represented by vertical bars that mark interquartile ranges of the vote distributions for every face. As expected, perceptual scaling for the newly added face (F1\*) was similar to that for other faces that matched the physical description of the perpetrator.

The relative voting score distributions for the remaining TA filler faces were also similar to those seen for the TP conditions (cf. Figs. 4a and 6). The correlation of voting scores for the five fillers that are common to both TP and TA conditions was high (0.86, $p = 0.036$). We observed only one significant scaling difference between voting scores in TP and TA conditions: face F4 was scaled higher than F2 in the TP condition and the reverse was true in the TA condition. In both conditions, F2 and F4 were the filler faces ranked least similar to (and thus least confusable with) the perpetrator.

**Signal detection analysis of recognition memory signals**. Our perceptual scaling data provide insights into the structure of recognition memory without asking each subject to make a unique choice and thus without engaging a decision criterion for identification. Using estimates of central tendency and variance obtained from perceptual scaling, identification is elegantly redefined as a problem of statistical inference. The distributions of votes cast for each lineup face (Figs. 4a and 6) can be used to estimate the likelihood of correct and incorrect identifications for a given decision criterion, using the tools of signal detection analysis[27–29]. (Identification here refers to classification of a given face as a target. Correct identifications are targets classified as targets; incorrect identifications are non-targets classified as targets.) We applied this method and assessed classification

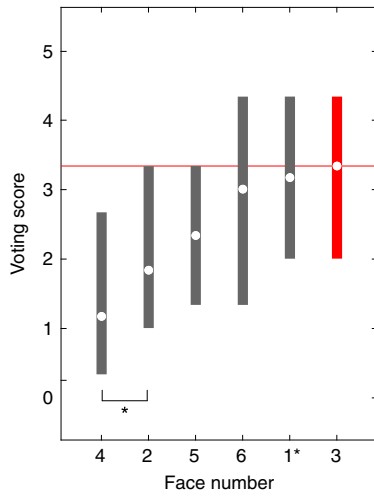

**Fig. 6 Results of perceptual scaling in the target-absent condition.** Structure of recognition memory is revealed by voting scores obtained from the paired comparison (PAR) target-absent lineup ($N = 70$ subjects). This figure has the same format as Fig. 4a with the exception that label '1' in the abscissa of Fig. 4a, referring to the target face in the target-present condition, is substituted by label '1\*', referring to the new face (F1\*) that replaced the target in the target-absent condition. Significant differences between the faces in adjacent ranks are indicated by asterisk ($p < 0.05$). A nonparametric statistical test was used to evaluate differences, as described in Methods (section Perceptual scaling of lineup faces). This is a one-sided statistical test.

performance for every decision criterion, thus deriving a criterion-independent measure of the ability of an optimal statistical classifier to make lineup identification decisions based on perceptual scaling data from a population of witnesses. Because of measurement uncertainty, performance of the optimal classifier will always be less than perfect, but it is the best performance that can be achieved with the information available.

The outcome of our signal detection analysis is conveyed in the form of ROC[27,29], which is a method to visualize relative probabilities of correct vs. incorrect identifications as a function of decision criterion. (The term "decision criterion" here refers to a statistical criterion value procedurally applied to overlapping voting score distributions, in accordance with its use in signal detection theory[27]. This is conceptually similar to, but should not be confused with, the recognition memory factor cited in the Introduction, in which decision criterion refers to the memory strength that a human observer requires to make an identification.) While ROC curves have been employed previously

to evaluate eyewitness performance with traditional SIM and SEQ lineups[8,10,13,30,31], our use of this approach is distinctive because it is applied to a complete set of recognition memory signals, derived from each subject in a manner that avoids decision bias. By contrast, traditional lineups must integrate decisions across subjects and often use expressed confidence as a proxy for decision criterion.

In the next section, we present ROC curves obtained from these analyses. These curves fall into two categories: those derived by averaging across the subject population and those reflecting individual subject performance.

**Population signal detection analysis.** Our population analyses are based upon data obtained from a large number of face pairings presented to each of many subjects. These analyses reveal general tendencies in the data in a form that facilitates comparison with traditional lineup studies, which necessarily yield population-based ROCs.

We first applied our signal detection analysis to perceptual scaling data from both TP and TA lineups. For the TP condition we determined for every decision criterion the probability of correctly classifying the target face as target. In the TA condition each of the six lineup faces was drawn from the same parent distribution, in the sense that they were all chosen to match the physical description of the perpetrator. For this reason, there was no single face designated as the innocent suspect; we considered the likelihood that any one of the faces in the TA lineup could be identified incorrectly. We thus determined separately for each face the probability (for each decision criterion) of incorrectly identifying that face. Following the same protocol used for this purpose in traditional SIM and SEQ lineups[8], the six face-specific probability measures were then averaged to yield the probability of incorrectly classifying a TA lineup face as target.

Correct and incorrect classification probabilities, computed as described above for each decision criterion, are plotted on the ordinate and abscissa of the ROC shown in Fig. 7. We call this curve a recognition ROC because it derives from recognition memory signals estimated by means of perceptual scaling. The area under the recognition ROC (area under the curve, AUC; 0.75, $p < 10^{-6}$) is a criterion-free index of the degree to which an optimal statistical classifier is capable of correctly distinguishing between the perpetrator and an innocent suspect based on scaling data from the entire subject population. The recognition ROC conveys the same performance metrics that are commonly reported in studies of traditional SIM and SEQ lineups, but does so by exploiting the full structure of recognition memory for lineup faces, while at the same time avoiding the ambiguity of witness' decision criteria for identification.

With these PAR lineup performance metrics in hand, we can compare the outcome of our approach to published data on performance obtained using traditional lineups. As noted above, we found our lineup to be unfair based on perceptual scaling, since there are significant differences between the responses elicited by our lineup faces (Figs. 4a and 6). We thus compared our performance data to that from a representative experiment using a lineup that was also acknowledged to be unfair (Experiment 2 of ref. [8]). The sequential and simultaneous lineup results from this earlier study are plotted together with our recognition ROC in Fig. 7. We recognize that there are procedural and stimulus variations between studies (such as degree of lineup fairness) that will affect discriminability. Nonetheless, the performance evinced by the PAR lineup in this inter-study comparison suggests that our approach has significant potential as a tool for eyewitness identification. An important goal of future studies will be to systematically vary lineup fairness (quantified by

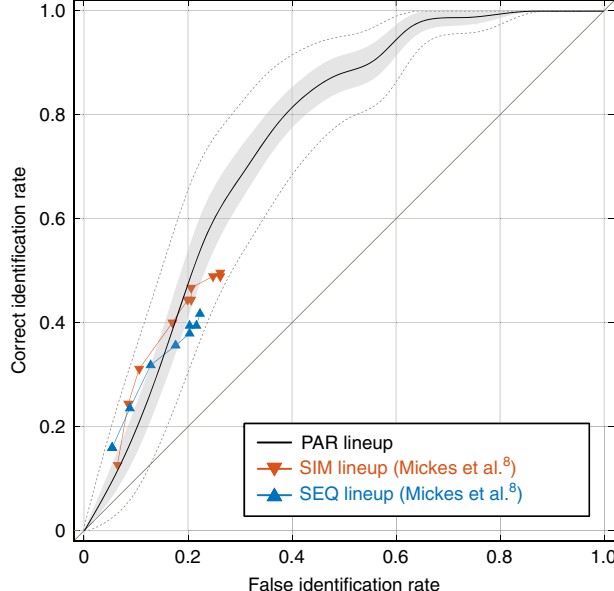

**Fig. 7 Signal detection analysis of recognition memory in the group of all subjects.** The black curve is a nonparametric estimate of the recognition receiver operating characteristic (recognition ROC) of collective subject performance in our paired comparison lineup (PAR). Correct identification rate, plotted on the ordinate, is derived from analysis of data obtained from the target-present condition ($N = 62$ subjects). False identification rate, plotted on the abscissa, is derived from analysis of data obtained from the target-absent condition ($N = 70$ subjects). Gray shaded region contains 50% error of the estimate (computed as described in Methods). Dashed lines bound a region that contains 95% error of the estimate (equivalent to the 95% confidence region; see Methods). Data plotted in orange (down-pointing triangles) and cyan (up-pointing triangles) are partial ROCs of collective subject performance in simultaneous (SIM) and sequential (SEQ) lineups obtained by Mickes et al.[8].

perceptual scaling) and thereby determine the true impact of this variable on performance.

**Individual signal detection analysis.** Because our perceptual scaling method also provides recognition memory metrics for individual subjects (e.g., see Fig. 5), we sought to apply the same signal detection method to evaluate single-subject performance. This is an unprecedented opportunity since traditional (SIM and SEQ) lineups do not allow one to assess individual eyewitness discriminability. However, individual subjects can only participate in the TP or the TA condition, but not both. In practice, this means that the probability of misidentifying an innocent suspect (which for population data was assessed using the TA lineup) must be estimated from the probability of identifying a non-target face in the TP lineup. This simulated target-absent approach is based on the premise that the probability of classifying a non-target face as target when viewing the TA lineup should, on average, be no different from the probability of classifying a non-target face as a target when viewing the TP lineup. Conceptual and empirical support for this premise is threefold:

(1) With the exception of the target face, all faces included in TA and TP lineups were drawn from the same parent distribution of faces, in the sense that they were all chosen to match the physical description of the perpetrator. Incorrect identification of a filler in the TP lineup should on average occur with the same likelihood as incorrect identification of any face in the TA lineup, provided that

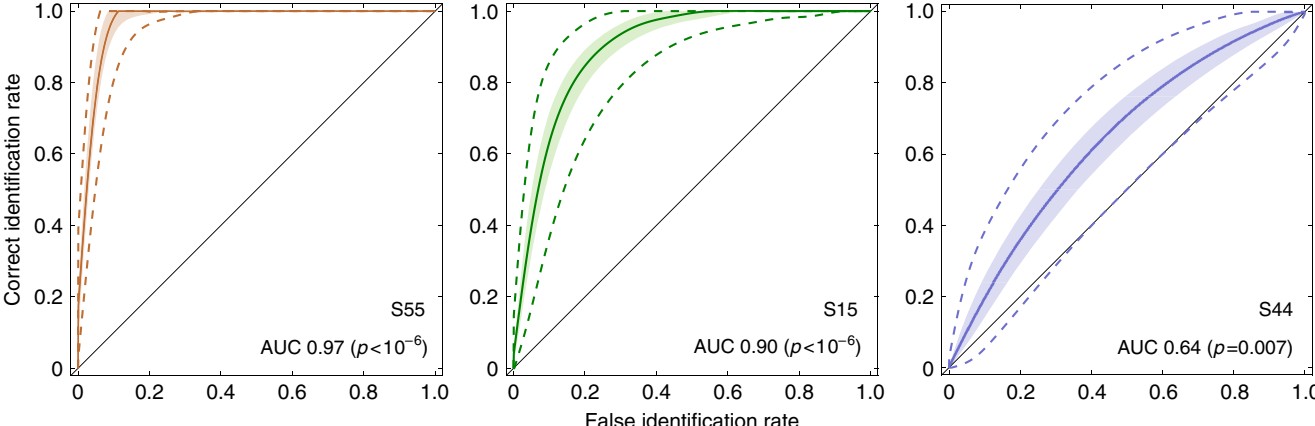

**Fig. 8 Signal detection analysis of individual recognition memory for three subjects.** Individual recognition ROCs are plotted for the three subjects whose voting functions appear in Fig. 5, matched by color and identified by subject number. Correct identification rate, plotted on the ordinate, is derived from analysis of data obtained from the target-present condition for the indicated subject. False identification rate, plotted on the abscissa, is derived from the simulated target-absent analysis (see Methods) of data obtained from the target-present condition for the indicated subject. As in Fig. 7, shaded regions in each panel contains 50% error of the estimate, and the dashed lines bound a region that contains 95% error of the estimate (equivalent to the 95% confidence region; Methods). Each panel also displays the subject identification number and an estimate of the area under the recognition ROC curve (AUC) at bottom right. The error of estimation of ROC and differences of measures of ROC from chance were computed as described in Methods (section Individual ROC analysis of eyewitness data).

the presence or absence of the target in the lineup does not alter the scaling of the remaining faces.

(2)  Well-developed arguments from voting theory (see Methods) predict that the presence or absence of the target in the lineup should not alter perceptual scaling of the remaining faces. Consistent with this prediction, the correlation of voting scores for the five non-target faces that are common to both TP and TA conditions was high (0.86; $p = 0.036$). We did observe one order reversal between the two conditions (cf. Figs. 4a and 6), which is a minor violation of the voting theory rule known as Limited Independence of Irrelevant Alternatives (see Methods). However, because the reversal occurred for two non-target faces (F2 and F4) that were the least confusable with the perpetrator, we believe that it does not invalidate our use of the simulated target-absent condition to compute individual ROCs.

(3)  Though impossible for individual subjects, our signal detection analysis can be performed using population data collected from both actual and simulated TA conditions in the PAR experiment. We found that the resulting recognition ROCs are similar: AUC values were 0.77 and 0.70 for actual and simulated TA conditions, respectively. This means that, on average, the probability of identifying a target relative to the probability of identifying a non-target is nearly the same for actual and simulated TA conditions. This population-level discovery supports our use of the simulated TA condition in the analysis of individual subject data. It furthermore suggests that the actual TA condition, which is a necessary feature of traditional SIM and SEQ lineup experiments, can be dispensed with when using the PAR procedure.

Results for individual subjects obtained using the simulated TA condition are illustrated in the form of recognition ROCs (Fig. 8) for the three subjects whose voting functions appear in Fig. 5. Each ROC curve summarizes the best possible discriminability of the target in our lineup relative to the average of all non-target faces. The area under the ROC curve (AUC) for each subject (displayed in each panel of Fig. 8) is thus a criterion-independent measure of individual subject performance. The confidence intervals associated with each curve can be used to statistically evaluate the relative certainty of each subject, and the degree to

which performance differs from chance. For example, discrimination performance of subject S55 (left panel in Fig. 8) is extraordinarily good. By contrast performance of subject S44 (right panel in Fig. 8) is barely distinguishable from chance ($p = 0.007$).

Measures of criterion-independent discriminability are not obtainable from individual subjects (nor from actual witnesses) using traditional lineups. Inferences about the correctness of identification in such lineups are often derived from witness statements of confidence, based on the observation that a confident witness is more likely to make an accurate identification[14,15]. The PAR lineup, by contrast, yields an objective quantitative index of performance, based on discriminability of recognition memory signals elicited by lineup faces. The PAR lineup can therefore be used to compare performance of individual eyewitnesses in a criterion-independent fashion. In particular, the method offers an objective performance criterion by which witnesses can be triaged based on their certainty.

## Discussion

The forgoing analyses suggest several key benefits of the PAR procedure relative to traditional SIM and SEQ lineups. All of these benefits are tied to the fact that the PAR procedure is not limited to a single report of identification, but rather renders rich quantitative detail about the structure of a witness' recognition memory for all lineup faces. Application of signal detection analysis to the scaled relationship between lineup faces and memory allows us to circumvent covert decision biases and render criterion-independent measures of discriminability. Results of our initial analysis of PAR population data are comparable with analyses of population data obtained using traditional lineups, though we stress the critical role of lineup fairness in making such comparisons. Indeed, one of the key features of our perceptual scaling approach is that it provides an immediate quantitative measure of lineup fairness, which we propose as a valuable tool for future studies of eyewitness performance.

Another important benefit of the PAR approach is that performance measures can be computed directly from the distributions of voting scores obtained using a single target-present lineup. We suggest that this new capability can eliminate the need (characteristic of traditional lineup procedures) for testing an

additional group of subjects who view only target-absent lineups in order to obtain a measure of incorrect identifications. Our findings support this proposal. Finally, our approach is not limited to population data analysis, as is the case for traditional procedures. Rather, the PAR lineup delivers a quantitative measure of performance based on individual eyewitnesses, which could be of enormous utility in assessing the value of individual eyewitness evidence.

For all of these reasons, we suggest that the PAR procedure may prove useful both as a research tool for evaluating effects of other variables on eyewitness performance and as a practical tool for investigation and prosecution of crimes. In a court of law, we envision that the PAR approach would augment direct testimony of an eyewitness with expert testimony on the classification of stimuli, as is routinely done in other forensic disciplines[32,33]. In this sense, using signal detection analysis to compare recognition memory signals is directly analogous, for example, to a forensic fingerprint examiner's use of standardized statistical methods for comparison of fingerprints[34].

Despite its many merits, there are potential concerns associated with the PAR procedure. For one, this procedure requires multiple viewings of each facial photograph, which could lead to new face memories that confound or compete with that of the perpetrator, and thus interfere with comparative judgments over the course of the lineup test. We believe such effects to be minimal or nonexistent because all faces appeared equally often in the PAR lineup and performance was comparable to that seen with traditional lineups.

A second potentially problematic feature of the PAR lineup is the fact that the witness never explicitly identifies the perpetrator. It is easy to imagine defense counsel's vociferous objection on these grounds. Rather than a scientific problem, this concern reflects a deep-seated cultural problem associated with our criminal justice system. The problem stems from naïveté about how people make decisions, which has long bedeviled eyewitness testimony and rules for reporting and testimony by forensic examiners. In particular, human decisions based on sensory information are necessarily probabilistic. A witness viewing a traditional SIM or SEQ lineup always confronts some uncertainty, but the decision is often rendered and interpreted as certain. Yet because the decision criterion employed by an eyewitness viewing a traditional lineup is unknown, and potentially influenced by undetermined perceptual, social or cultural factors, we have no immediate knowledge of the amount of uncertainty that the witness was willing to bear in making an identification. In other words, the longstanding practice of encouraging witnesses to say "he's the one" is where the problem lies.

The PAR lineup offers a practical solution to this cultural problem. It cannot overcome the limits imposed by uncertainty, but it acknowledges this uncertainty and offers a quantitative index of it. A jury presented with this numerical evidence is in a more informed position to determine guilt, as compared to a jury presented with the faux certainty of a traditional lineup identification.

## Methods

**Experimental design**. The experiment consisted of two parts. In the first part, every subject was presented with the same video recording of a mock crime: a 42-s fragment from the theatrical film God Bless America (Robert Goldthwait, 2011). In the second part, conducted on the following day, each subject was presented with one of three lineup types. Subjects viewed the same six lineup faces, regardless of lineup type. One of the lineup faces was that of the actor who played the role of perpetrator in the crime video. This face was designated the target and lineups that included this face were termed target-present. The other lineup faces were fillers selected based on certain attributes (race, facial hair, etc.) that matched the description of the perpetrator. In all three lineup types, subjects were told that the target may or may not be present in the lineup. As noted below, some subjects in

the PAR lineup condition viewed a target-absent lineup, in which the target face was replaced with another face that matched the description of the perpetrator.

**Lineup types**. The simultaneous (SIM) lineup (Fig. 1a) consisted of a single presentation of six head-and-neck images (face images), with the spatial order of images randomized between subjects. Subjects used the computer mouse to either select one of the images, indicating that the face matched their memory of the perpetrator, or select the option that none of the faces matched their memory.

The sequential (SEQ) lineup (Fig. 1b) consisted of sequential presentation of the six face images, one image at a time, with the temporal order of images randomized between subjects. Upon viewing each image, subjects used the computer mouse to select whether or not this face image matched their memory of the perpetrator. Subjects were pre-informed that only their first choice was counted as identification[7].

The paired comparison (PAR) lineup (Fig. 1c) consisted of 15 possible pairings (pairwise combinations) of the six face images. Each pair was presented three times, for a total of 45 trials. The order of image pairs across trials, and the left/right order of images within a trial, was randomized between subjects. Upon viewing a pair of images, subjects were required to use the computer mouse to select the image that was more similar to their memory of the perpetrator (forced choice) thus yielding a single vote for the selected image.

Traditional simultaneous and sequential lineups were included in this study solely for the purpose of comparing the relative frequencies of top-ranked faces (Fig. 3) across the different lineup conditions (PAR, SIM and SEQ). All subsequent procedures, including perceptual scaling and ROC analyses, were employed to characterize the application and utility of the PAR lineup.

**Conditions of PAR lineup composition**. We applied the PAR lineup procedure in two experimental conditions: target-present (TP) and target-absent (TA). The TP lineup was composed of the target face and five fillers that matched the description of the perpetrator. The TA lineup was identical to the target-present lineup, with the exception that the target face was replaced with another face that matched the description of the perpetrator.

**Lineup rejection**. A lineup was "rejected" when the witness failed to identify a single face in the lineup. In our study, SIM and SEQ lineups were rejected when the subject did not explicitly select a face. The PAR lineup was rejected when the voting scores of two or more face images tied for the top rank.

**PAR lineup data analysis**. We employed two established experimental techniques to analyze data from PAR lineups: perceptual scaling and signal detection analysis. The former allowed us to estimate the recognition memory signals elicited by each lineup face and the latter enabled us to determine the best possible discriminability of the perpetrator relative to an innocent suspect.

**Perceptual scaling of lineup faces**. The complete set of votes collected from each subject was analyzed as shown in Fig. 2: Responses were first organized into a voting matrix (Fig. 2a), from which the six voting scores and voting function were derived (Fig. 2c–e). Linear regression was used to determine the slope of the voting function. The voting score distribution for each lineup face is represented in Fig. 4 by the medians and interquartile ranges.

To evaluate whether differences between voting scores for different face images were statistically significant, we performed bootstrap analysis of the voting score distributions in every subject group (all subjects, correct subjects, incorrect subjects). The probability of obtaining a difference of voting scores by chance was estimated by subtracting the higher-scores distribution from the lower-scores distribution and computing the fraction of negative values of the difference.

**Signal detection analysis of discriminability of lineup faces**. We assessed discriminability of the perpetrator vs. an innocent suspect using a standard process of statistical inference. As noted in the main text, voting score distributions serve as estimates of the recognition memory signals elicited by lineup faces. These distributions were used to compute the relative probabilities of correctly classifying the target as target (correct identification rate) and incorrectly classifying a non-target as target (false identification rate), as a function of decision criterion.

**Receiver operating characteristics**. The results of our signal detection analyses are conveyed in the form of receiver operating characteristic (ROC) curves, which plot as a function of decision criterion the probability of correctly classifying the perpetrator vs. the probability of incorrectly classifying an innocent suspect. Signal detection analyses were carried out and ROC curves were produced separately for population and for individual subject data.

**Population ROC analysis of eyewitness data**. Using data from the aforementioned target-present and target-absent conditions, we determined the population ROC curve by standard methods[27,29], which is by moving the decision criterion along the dimension underlying the voting score distributions and computing

corresponding classification probabilities for the perpetrator (designated as hit rate in the signal detection literature) and for innocent suspects (false alarm rate). Hit rate was estimated by computing the fraction of correct identifications of the target under each decision criterion in the population distribution of votes in the TP condition. False alarm rate was estimated by computing the fraction of incorrect identification of a non-target under each decision criterion in the population distribution of votes in the TA condition. These classification probabilities are integrated into the area under the ROC curve (AUC), which thus serves as a criterion-independent index of the degree to which an optimal classifier can correctly distinguish target and non-target faces based on the distributions of population scores.

We determined the error of estimating the population ROC curve using bootstrap analysis. We resampled the voting scores making up the two distributions used to derive the ROC: one for the target in the TP condition and the other for non-targets in the TA condition. On each cycle of resampling, we derived a new ROC (resampled ROC) and computed its AUC. This way we obtained a distribution of resampled ROC curves and a distribution of resampled AUC indices. We used the distribution of resampled ROC curves to determine the 50 and 95% regions of error of ROC marked in Fig. 7. We used the distribution of resampled AUC indices to evaluate the probability that our results could be obtained by chance.

**Individual ROC analysis of eyewitness data.** As described in the main text, the voting scores for individual subjects can be used to estimate the means of recognition memory signals elicited by lineup faces, and the slope of the voting function can be used to quantify the variance associated with the recognition signals. From these statistical moments it is possible to compute classification probabilities for target and non-target faces.

As noted above, we derived the population ROC using the TP condition to determine the probability of correctly identifying the target, and using the TA condition to determine the probability of incorrectly identifying a non-target. Individual subjects, however, can only participate in either TP or TA conditions. We therefore derived single-subject ROCs using data from the TP condition to determine probability of correct identification, as before, but using a simulated TA condition to determine the probability of incorrect identification. In this simulated TA condition, the target face was removed from analysis. Removal of the target face is legitimate for this purpose if it does not alter scaling of the remaining lineup faces. Statistical decision theory has shown that removal of a voting option (the target in this case) from consideration is legitimate if it satisfies the stringent mathematical criterion of local independence of irrelevant alternatives (LIIA), in which scaling is robust over removal of the top vote-getter[35–39]. Paired comparison scaling procedures are among the few methods known to satisfy this LIIA requirement[40,41]. In other words, the outcome of a lineup with five non-targets should be unaffected by the presence or absence of the target (the top vote-getter in the population average). It follows that voting score distributions obtained from those five non-targets should enable us to evaluate the probability that a non-target is wrongly classified as target. This argument, together with supporting results summarized in the section "Individual subject analysis of eyewitness data" in the main text, justify our approach to derivation of individual ROC.

This individual subject procedure is analogous to that used for population data, with two exceptions: first, that a simulated TA condition was used to estimate false alarm rates (as we explain in the main text); second, that the following procedure was used to compute mean and variance of voting scores for each face:

*Step 1.* We set the means of the probability distributions $\mu_i$, $i = \{1...6\}$ to the subject's six measured voting scores and assumed that (a) the variances $\sigma^2$ of these distributions were the same for every face; (b) the distributions were normal, $N(\mu, \sigma^2)$. We estimated each subject's recognition memory variance ($\sigma^2$) in the next step.

*Step 2.* We estimated each subject's recognition memory variance ($\sigma^2$) by matching the slope of the voting function produced by the model (defined in Step 1) to the slope of the measured voting function, as follows. Using the subject's model, we simulated the PAR experiment assuming different values of $\sigma$. For every tested value of $\sigma$, we simulated the comparisons among 45 pairs of images. For each pair, we sampled one response from each of the two simulated distributions that corresponded to two images in the pair. The image whose sample had a larger value was judged as more similar to the target. By this process, we derived the simulated voting matrix and voting function for the subject, following the procedure described in Fig. 2. We repeated this procedure for multiple values of $\sigma$, and found the value $\sigma^*$ for which the slope of the simulated voting function was most similar to the slope of the measured voting function for this subject. Hence, $\sigma^{*2}$ was an estimate of the subject's recognition memory variance.

*Step 3.* As for population data, the target-present and the (simulated) target-absent analyses were applied to voting score distributions in order to determine, respectively, the probability that the target is correctly identified as target and the probability that a non-target is incorrectly identified as target. (Distributions for all fillers in the simulated target-absent condition were used to determine the false identification rate.) These correct and incorrect probabilities were computed for each decision criterion and plotted as ROC curves. The AUC for each curve serves as a criterion-independent index of the degree to which the optimal classifier can correctly discriminate target and innocent faces based on the voting score distributions obtained from a single witness. As such, the

individual-witness AUC is a quantitative index of individual witness performance.

Similar to our analysis of population ROC curves, we used the distribution of individual ROC curves produced at Step 3 of the above procedure to determine the 50 and 95% regions of error of ROC marked in Fig. 8. These regions of error allow one, first, to evaluate whether performance of individual eyewitnesses is different from chance and, second, to evaluate the degree to which ROC curves of individual eyewitnesses are different from one another.

**Apparatus.** Mock crime and lineup face images were viewed in a quiet light-tight behavioral testing room. All visual stimuli were rendered in color on a computer monitor (Sony Trinitron Multiscan 500PS, resolution 1024 × 768 pixels, 60 Hz, 32-bit True Color) and viewed from a distance of 62 cm using a chin rest. The video recording had the resolution of 1280 × 720 pixels at the screen size of 40 × 18 cm (35.8 × 16.5 degrees of visual angle) and rendered at the rate of 25 frames/s. Face image resolution was 167 × 167 pixels. Each face image had a screen size of 7 × 7 cm (6.5 × 6.5 degrees of visual angle). Experiments were controlled using the software platform MATLAB, Release 2017b, The MathWorks, Inc., Natick, Massachusetts, USA.

**Participants.** Subjects were primarily (96%) undergraduate students from the University of California, San Diego, who received course credit for their participation. A small number of subjects (4%) were college-age individuals from the La Jolla community, who received monetary compensation for their participation. Two hundred and two (202) subjects participated in the experiments (mean age 20.7 years, standard deviation 2.9 years; 67% female): 62 subjects were presented with the PAR target-present lineup and 70 subjects were presented with the PAR target-absent lineup; 33 subjects with the SIM lineup, and 37 subjects with the SEQ lineup. All subjects had normal or corrected-to-normal visual acuity.

**Human subject protection.** In accordance with ethical standards set by US laws and regulations, we protected the welfare, rights, and privacy of human subjects who participated in these experiments. Subjects were informed of their rights as experimental participants and they provided written consent. The human subject protocol was reviewed and approved by the Human Subjects Institutional Review Board of the Salk Institute for Biological Studies (protocol #17-0002).

**Reporting summary.** Further information on research design is available in the Nature Research Reporting Summary linked to this article.

## Data availability

The data that support the findings of this study are available on the Open Science Framework in the listing for the following research project: "Enhancing Eyewitness Performance by Optimizing Context." In particular, the source data underlying Figs. 2–8 are available under the project component Data at https://osf.io/n7vme/. A reporting summary for this Article is available as a Supplementary Information file.

## Code availability

Data were collected and analyzed using the commercial software platform MATLAB, Release 2017b, The MathWorks, Inc., Natick, Massachusetts, USA.

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

## Acknowledgements
We wish to thank numerous colleagues from both science and criminal justice communities who provided advice and encouragement. We are particularly indebted to John T. Wixted for sharing his deep insights into the problem of eyewitness identification and his enthusiasm for our approach to the problem. We also thank Brent M. Wilson for many clarifying discussions and David T. Van for assistance with data collection and analysis. This work was supported by awards from Arnold Ventures and from the Innovation Grants Program of the Salk Institute for Biological Studies.

## Author contributions
T.D.A. and S.G. conceived of the idea, developed the hypotheses, and designed the experiments and methods of data analysis. D.D., F.H., and Y.W. collected the data. S.G., F.H., and Y.W. analyzed the data. S.G. and F.H. conducted modeling of individual eyewitness performance. T.D.A. and S.G. wrote the paper.

## Competing interests
The authors declare no competing interests.
