## [Peer Review File · Nature Communications]

Reviewers' comments:

Reviewer #1 (Remarks to the Author):

In a standard lineup procedure in the US, eyewitness try to pick the target (the guilty suspect) from an array of pictures of the suspect and fillers. The suspect is either innocent or guilty. And the fillers are known to be innocent. The pictures are either shown all together, which is known as a simultaneous lineup, or one at a time, which is known as a sequential lineup. The authors tested a new lineup, the paired comparison (PAR) lineup. In the PAR, eyewitness participants are shown a series of two pictures of the lineup member and pick which is more familiar of the pair.

The motivation for designing the PAR lineup arose from concerns about response bias and memory strength. The authors wrote, "Traditional lineup procedures rely on overt eyewitness responses that confound two covert factors: the strength of recognition memory and the criterion for deciding what perceptual evidence is sufficient for reporting a match." This is an issue when the diagnosticity ratio is used, however, ROC analysis measures discriminability independently from response bias (e.g., Rotello & Chen, 2016; Wixted & Mickes, 2012). The authors should consider acknowledging this.

The best lineup procedure is the one that gives rise to more correct identifications *and* fewer false identifications than other lineups (i.e., better ability to discriminate innocent from guilty suspects). For this new lineup to be considered a viable alternative to the standard lineups, PAR needs to be tested with innocent suspects too. ROC analysis was conducted by plotting correct ID rates and filler ID rates (using IDs of fillers from the target-present lineups), but that doesn't tell us what we need to know. We need to know if this procedure reduces innocent suspect IDs (in addition to increasing guilty suspect IDs), so need to real false alarm rate (the rate of identifying the innocent suspect).

The experiment needs to be done with target-absent lineups too, then the discriminability across lineup types can be compared. As it stands, only correct IDs are provided. There were many correct IDs with PAR, fewer with simultaneous lineups, and even fewer with sequential lineups. But providing this information without providing false IDs, and conducting ROC analysis, we can't actually tell if this new lineup procedure should be considered for use in the real world. I hope they do this, I'd love to know the answer.

Reviewer #2 (Remarks to the Author):

This review is going to be relatively short because there is a design flaw that is so fundamental that, frankly, these data cannot be salvaged for purposes of making a meaningful and defensible argument

about improving eyewitness identification.

Before I get to the critical design flaw, I should note that the authors are apparently unaware that this approach of using a continuous measure is definitely not new. Whether one uses similarity ratings or confidence, there are well-designed experiments that examine eyewitness performance that do not ask the witness to make a yes/no type decision. The work of Neil Brewer and his lab is a dominant example. And, the work of Fitzgerald & Price (2016) and more recently Bruer and Price (2017) uses a so-called “face off” procedure that involves pairwise comparisons. So, there is a literature out there that is highly similar to this so-called “novel” procedure and any eyewitness scientist who knows the literature knows that this is not so novel.

But the real problem with this experiment, which cannot be fixed by rewriting the manuscript, is that the lineups used in this experiment did not target-absent lineups. It has been known for almost 40 years (since Lindsay and Wells, 1980) that in order to test the accuracy of eyewitness identification from lineups you have to have both target-present and target-absent conditions. And the authors cited Lindsay and Wells (1985) that makes this point very clearly. There is no debate about this; any experiment comparing one lineup procedure to another must include conditions in which the target is absent as well as conditions in which the target is present. It is as if the authors had not fully read the articles that they have cited. For example, the authors cited Wixted and Wells (2017), which clearly explains that comparing one lineup procedure to another requires that the witness be randomly assigned to a lineup that either does nor does not include the target. That same lesson is apparent in at least 12 of the articles that the current authors cited.

I do not think it is incumbent on me as a reviewer to explain the logic of why researchers have to manipulate the presence versus absence of the target in a lineup (as a factor in the design) in order to test eyewitness identification performance. But here is a short version of the argument: A police lineup contains one (and only one) suspect. Everyone else is a known-innocent filler. The suspect might or might not be the culprit. What we need to know in estimating performance in an experiment is how often the witness picks the suspect when the suspect is the culprit and how often the witness picks the suspect when the suspect is innocent. You can only test the latter when the lineup does NOT contain the target. Hence, every test that compares one lineup procedure to another lineup procedure MUST include both target-present and target-absent trials. After all, one lineup procedure might be superior when the culprit is present and the other superior when the culprit is absent. In fact, that is a common finding,

The bottom line is that this experiment would never be accepted by eyewitness identification scientists as a test of which lineup procedure is superior because it fails to include the target-absent conditions. So, even if this were a totally novel procedure, which it is not, this work would not be acceptable as evidence of which procedure is superior.

Reviewer #3 (Remarks to the Author):

This is the first review of 'Perceptual scaling improves eyewitness identification' by Gepshtein and colleagues. The report examines face recognition memory in a lab analogue of an eyewitness lineup procedure, comparing three procedures. The novel procedure introduced by the researchers involves the method of paired comparison, a psychophysical discrimination procedure traced back to Thurstone, which in principle allows one to place each stimulus on an evidence continuum. The procedure involves having the S select each of all possible pairwise combinations of the lineup members (1 suspect and 5 fillers) is most like their memory of the perpetrator; 15 possible pairs). The authors demonstrate several interesting characteristics of the procedure and suggest it is superior to simultaneous and sequential procedures because it is free of a particular type of decision criterion (namely the criterion to make an identification). The report is interesting and clearly written and I think it makes a good case for further examination of the method of paired comparison in this domain. However, I'm not sure the authors have established it is superior to the other methods since it isn't really formally compared in any manner I can tell. Moreover, I think there are several interpretive and technical issues not adequately addressed in the current version. I outline these concerns in more detail below followed by a summary paragraph.

MAIN CONCERNS

I found the characterization of decision models of recognition confusing. The authors continually refer to the 'implicit response' and 'implicit decision criterion' of subjects. I think the former is referring to the internal evidence upon which decisions are based. If so, 'implicit response' is confusing for two reasons. First, recognition memory evidence, or at least one component of recognition memory evidence, is assumed explicit, not implicit (viz., contextual recollection). Indeed, I believe the authors use the term 'declarative memory' in the text which, of course, assumes subjects can report the contents of memory in an overt fashion. The second aspect of 'implicit response' that is confusing, if the authors are referring to recognition evidence, is that 'response' implies a decision that has been rendered. However, under statistical decision models the evidence doesn't dictate the form or nature of the decision, which instead reflects a decision operation that is often modeled to optimize, in some statistical fashion, the translation of evidence into judgment. There are similar problems with the notion of an 'implicit decision criterion.' Decision criteria or thresholds are usually assumed to be at least partially under subject control, which is why subjects can adopt a more or less conservative approach when instructed or given payout matrices. For both of these phrases, the authors may be mixing the idea that the decision and evidence process are unknown/covert/internal, with the idea they are implicit. The latter traditionally is taken to mean unconscious and involuntary and I don't think this is the authors' intent but I'm not certain.

While I find the method of paired comparison interesting, it might be helpful if the authors could explain a little more concretely how the performance can be taken to illustrate the variance of targets and lures. If I understand the methods section correctly, it is necessary to assume that the target and lures have equal variance in order to use the method to place the stimuli along a single dimension. If so, then this is a potential problem because the single item, confidence based recognition ROC is inconsistent with the equal variance assumption, leading to the unequal variance signal detection model (e.g., Mickes, Wixted

& Wais 2007), and/or dual process models (e.g., Yonelinas 1994; Mickes & Wixted 2010) that do not make this assumption. Of course, if the assumption of equal variance isn't valid, then the scaling solution the method of paired comparison produces in this domain is spurious.

The instructions given to the participants are not covered in sufficient detail. Currently, it is unclear what they were led to believe about the presence of the perpetrator in the lineup procedure. This is critical since in both the sequential and simultaneous procedures subjects can reject the lineup. In contrast, the PAR method they cannot. Given that the discrimination performance of the simultaneous and sequential group depends upon beliefs about the prior probability the perpetrator is present (ideally), it would be useful to compare the methods across a range of biases regarding the likelihood the perpetrator is present in the lineup. Regardless, providing the exact instructions given to all three groups seems important.

The questions/hypotheses posed by the researchers at the bottom of page 2 are potentially undermotivated or not possible of being addressed by the current design. The first question is whether the PAR method provides useful information about the identity of the culprit. However, I fail to see how it cannot unless the observers had no memory of the culprit. Analogously, it would be odd to ask if the sequential procedure provided useful information about the identity of the culprit when it was introduced, since again, how could it not if observers have some memory for the culprit. The second question was whether the PAR results allow one to draw inferences about the utility of that witnesses testimony. I wasn't entirely certain how this question differed from the first, but I believe the authors are highlighting that the procedure may give a bias free estimate of accuracy for each individual. I think this is a notable goal and would be an advance over methods that allow subjects to reject the lineup as a whole, but presently it is not clear if this is what the authors are asking here. The final question addresses the practical benefit of the procedure. However, the paper doesn't seem designed to address this issue since it doesn't establish that the procedure is in some way clearly superior to the other two methods currently in use.

The paper sometimes conducts statistical tests that seem unnecessary or statistical comparisons of non-comparable data. For example, the authors note that the suspect was the top ranked choice in the PAR procedure, but this would necessarily be true unless memory was non-existent and it doesn't seem to require a statistical test. Perhaps more problematic is that the authors test the number of rejections of the lineup across the three procedures. However, these data are non-comparable because rejections in the PAR procedure have nothing to do with observers deciding that they do not want to risk identifying an innocent suspect...they must select a member of the pair on each trial. Instead, it appears that 'rejections' here reflect subjects for which two or more probes have the top rank in terms of selections. These different rates strike me as simply different things, and they are clearly calculated fundamentally differently. Given this, the statistical test strikes me as uninformative.

Interestingly, the authors demonstrate that across subjects who rank the filler highest, the target nonetheless has the highest mean rank. I believe this simply means that the lineup is fair among the fillers such that errors do not arise because one is preferentially similar to the suspect (i.e., a patsy).

While this is an interesting characteristic, it would presumably also arise for the other procedures if the observers were forced to rank the lineup members. Namely, that when averaged across those who ranked a lure as highest, the sample mean of the target would nonetheless be higher (i.e., an item analysis would show the highest mean for the target). Finally, the second-choice procedure in forced-choice procedures seems germane. Under this approach, during say four alternative forced choice recognition, observers are required to make a second choice after their first selection, corresponding to their guess if their first response were incorrect. Here, second choice accuracy is higher than chance, which presumably means that when averaged across subjects, the target in this sub-sample nonetheless has the highest item mean (e.g., Parks & Yonelinas 2009).

The authors perform a statistical comparison across a population ROC representing discrimination of the target face from all others (F1 vs \sim F1) versus an ROC with classes of the second most selected face (F3) and all others (except F1 and F3). I'm not sure what this test answers that is not already clear simply by directly looking at the ranks or selection rates the authors already show in Figure 3D. In other words, this seems like a pretty complex way to ask whether or not the subjects select the target more often than the next most highly selected item, but perhaps I am missing why ROCs are uniquely suited to answer this question.

As noted above, the PAR method seems to require that one assume target and lure evidence distributions are equally variable; an assumption that seems unwarranted given the recognition ROC literature. This means that one of the potential strengths of the procedure, the ability to provide bias free subject accuracy measures (Figure 4B) is undercut to the degree this assumption is critical. Is there some way the authors can test how critical this assumption is? For example, does the rank ordering of the individuals, in terms of AUC, change appreciably when the target distribution is assumed up to 1.5 times more variable than the lures? I imagine that if a simulation demonstrated that the rank ordering of AUCs across individuals was remarkably stable when target variability was randomized within a reasonable range (given the prior literature) it would be quite newsworthy. Conversely, if the rank ordering of AUCs heavily depends on the equal variance assumption (or the equivalence of the target variance across participants) then value of the AUC estimation would be undercut.

MINOR CONCERNS

In the Discussion the authors discuss confidence during simultaneous and sequential procedures noting confidence is collected 'based on the assumption that a confident witness is more likely to make an accurate identification.' However, this is not really an assumption, but an established fact. As a whole, more confident judgments are more accurate than less confident judgments in this literature and as Wells, Wixted, and Roediger have noted (e.g., Wixted & Wells 2017), accuracy conditioned on rendering the highest possible confidence, immediately after selection, is quite high...and invariably higher than accuracy when confidence is lower.

The fact the PAR doesn't require a target-absent lineup is sort of buried and not mentioned until the Discussion. This is a really neat feature although I'm a little unclear on its theoretical significance. Regardless, it is presumably linked to the forced choice nature of the procedure which, like traditional forced choice designs, eliminates the distinction between a false alarm and a miss. It is this aspect that

seems a little underemphasized in the Introduction. Namely, that decision modelers have struggled to deal with how to model the decisions in designs that allow the subject to reject the lineup for the simple fact that rejecting the lineup really doesn't map to the quality of the memory evidence of observers in any clear way. Some rejectors presumably may have pretty good signal to noise for the target and some may not. The PAR procedure seems to offer a way to force selection, eliminating this thorny modeling issue, but the researchers have not clearly demonstrated that it doesn't result in increased jeopardy for innocent suspects. I imagine that if they could demonstrate this (and that the rank ordering of individual accuracy estimates is not particularly dependent upon the equal variance assumption) then they would have a pretty influential study.

One of the individual subject ROCs doesn't appear to use 15 criteria (black) although the authors indicate that 15 thresholds were used. Additionally, it wasn't quite clear how the 15 were determined.

Is there some reason the authors use twice the number of subjects for the PAR as the simultaneous and sequential procedures?

Why not use confidence during the PAR? I don't see any reason why this would contaminate the procedure and it might allow a method of comparison to the other procedures.

SUMMARY

The authors present an interesting and novel approach to characterizing eyewitness accuracy. At present, the findings certainly suggest that the method of paired comparison warrants further research. However, the manuscript implies the method is manifestly superior to present approaches using confidence and yet there is nothing in the current report that directly demonstrates the method produces greater identification than say restricting decisions (in the simultaneous and sequential procedures) to the highest confidence category in the extant methods. Moreover, it is unclear how critical the equal variance assumption is to the modeling of individual subject ROCs. If the authors can demonstrate it isn't critical, that the method is relatively impervious to suggestive techniques, and entails less jeopardy to innocent suspects I think it would be quite newsworthy. I realize that is a tall order and perhaps not feasible in this short format. However, at a minimum it seems important to demonstrate that the individual ROCs are largely impervious to unequal variance possibilities and that, somehow, the procedure can more easily limit false identifications without the vagaries of subjective confidence inherent in the other approaches.

RESPONSE TO REVIEWERS

We are grateful for the time and expertise that the reviewers have brought to our manuscript. In the following, we present our responses to each critical comment or question made by these reviewers. Because there is a large number of such comments and questions and there is a considerable degree of content overlap between them, we have organized our responses by theme. There are 18 of these themes. Each is introduced with a statement or question (red font) that is intended to broadly capture what we understood to be the concern, followed by quoted text from the reviewers in question (small italic font), and then by our response. Although this approach involved shuffling reviewer text around a bit, we have addressed every substantive concern raised by the reviewers.

We recognize that the concepts and methods we have employed in this study are neither standard in the field of eyewitness identification (ID) research nor widely familiar to many working on this topic. As we summarize below and in the revised manuscript, our approach affords numerous advantages that are worth consideration for their potential both as a research tool and as an aid to criminal investigation. We do not maintain that this report is an ultimate statement on the matter. There is surely more work to be done. We offer this report for publication because it describes a truly novel approach to the problem and because that approach allows one to make informed decisions based on a holy grail of perceptual psychology, which is the scaled relationship between stimulus and mind.

We also note here that we have made a number of broad enhancements to the manuscript – in terms of narrative organization, consistency of terminology, content, and title – that were inspired by points of interest or confusion highlighted by the reviewers. The data and our main conclusions are the same, but we feel that the overall manuscript is vastly improved.

1) The most significant concern raised by reviewers is in regard to the degree to which the PAR lineup represents an advance over traditional lineups. There are two interrelated components to this concern: (a) The report fails to clearly identify in what sense the PAR procedure is “superior” to traditional simultaneous and sequential lineups, and (b) The report cannot possibly support claims of superiority in any case, because the method does not include target-absent lineups. In the following, we first cite specific comments made by the reviewers that state each of these components. This is followed by our response.

(a) Case for superiority of PAR lineup is unclear.

REVIEWER 1:

We need to know if this procedure reduces innocent suspect IDs (in addition to increasing guilty suspect IDs) [relative to traditional simultaneous and sequential lineups].

REVIEWER 2:

[Because it fails to include target-absent lineups] this work would not be acceptable as evidence of which procedure is superior.

REVIEWER 3:

The authors demonstrate several interesting characteristics of the procedure and suggest it is superior to simultaneous and sequential procedures because it is free of a particular type of decision criterion (namely the criterion to make an identification).

...The report is interesting and clearly written and I think it makes a good case for further examination of the method of

paired comparison in this domain. However, I'm not sure the authors have established it is superior to the other methods since it isn't really formally compared in any manner I can tell.

...The final question addresses the practical benefit of the procedure. However, the paper doesn't seem designed to address this issue since it doesn't establish that the procedure is in some way clearly superior to the other two methods currently in use.

... the manuscript implies the method is manifestly superior to present approaches using confidence and yet is there is nothing in the current report that directly demonstrates the method produces greater identification than say restricting decisions (in the simultaneous and sequential procedures) to the highest confidence category in the extant methods.

(b) Demonstration of superiority is impossible because target-absent conditions are not included.

REVIEWER 1:

*The best lineup procedure is the one that gives rise to more correct identifications *and* fewer false identifications than other lineups (i.e., better ability to discriminate innocent from guilty suspects). For this new lineup to be considered a viable alternative to the standard lineups, PAR needs to be tested with innocent suspects too. ROC analysis was conducted by plotting correct ID rates and filler ID rates (using IDs of fillers from the target-present lineups), but that doesn't tell us what we need to know. We need to know if this procedure reduces innocent suspect IDs (in addition to increasing guilty suspect IDs), so need to real false alarm rate (the rate of identifying the innocent suspect).*

The experiment needs to be done with target-absent lineups too, then the discriminability across lineup types can be compared. As it stands, only correct IDs are provided. There were many correct IDs with PAR, fewer with simultaneous lineups, and even fewer with sequential lineups. But providing this information without providing false IDs, and conducting ROC analysis, we can't actually tell if this new lineup procedure should be considered for use in the real world. I hope they do this, I'd love to know the answer.

REVIEWER 2:

...there is a design flaw that is so fundamental that, frankly, these data cannot be salvaged for purposes of making a meaningful and defensible argument about improving eyewitness identification.... the real problem with this experiment, which cannot be fixed by rewriting the manuscript, is that the lineups used in this experiment did not target-absent lineups. It has been known for almost 40 years (since Lindsay and Wells, 1980) that in order to test the accuracy of eyewitness identification from lineups you have to have both target-present and target-absent conditions. And the authors cited Lindsay and Wells (1985) that makes this point very clearly. There is no debate about this; any experiment comparing one lineup; procedure to another must include conditions in which the target is absent as well as conditions in which the target is present. It is as if the authors had not fully read the articles that they have cited. For example, the authors cited Wixted and Wells (2017), which clearly explains that comparing one lineup procedure to another requires that the witness be randomly assigned to a lineup that either does nor does not include the target. That same lesson is apparent in at least 12 of the articles that the current authors cited.

I do not think it is incumbent on me as a reviewer to explain the logic of why researchers have to manipulate the presence versus absence of the target in a lineup (as a factor in the design) in order to test eyewitness identification performance. But here is a short version of the argument: A police lineup contains one (and only one) suspect. Everyone else is a known-innocent filler. The suspect might or might not be the culprit. What we need to know in estimating performance in an experiment is how often the witness picks the suspect when the suspect is the culprit and how often the witness picks the suspect when the suspect is innocent. You can only test the latter when the lineup does NOT contain the target. Hence, every test that compares one lineup procedure to another lineup procedure MUST include both target-present and target-absent trials. After all, one lineup procedure might be superior when the culprit is present and the other superior when the culprit is absent. In fact, that is a common finding,

The bottom line is that this experiment would never be accepted by eyewitness identification scientists as a test of which lineup procedure is superior because it fails to include the target-absent conditions. So, even if this were a totally novel procedure, which it is not, this work would not be acceptable as evidence of which procedure is superior.

REVIEWER 3 (“MINOR”):

The fact the PAR doesn't require a target-absent lineup is sort of buried and not mentioned until the Discussion. This is a really neat feature although I'm a little unclear on its theoretical significance.

Regardless, it [the fact that the PAR doesn't require a target-absent lineup] is presumably linked to the forced choice nature of the procedure which, like traditional forced choice designs, eliminates the distinction between a false alarm and a miss. It is this aspect that seems a little underemphasized in the Introduction. Namely, that decision modelers have struggled to deal with how to model the decisions in designs that allow the subject to reject the lineup for the simple fact that rejecting the lineup really doesn't map to the quality of the memory evidence of observers in any clear way. Some rejectors presumably may have pretty good signal to noise for the target and some may not. The PAR procedure seems to offer a way to force selection, eliminating this thorny modeling issue...

AUTHORS' RESPONSE:

Our goal in this manuscript is to highlight numerous advantages afforded by the novel lineup approach we have introduced. Though one might argue (as we have done) that these advantages are collectively evidence of improvement over simultaneous and sequential procedures, the definition of improvement (“superiority”) used by the reviewers is explicitly the extent to which a lineup is more likely to yield correct IDs of targets vs false IDs of innocent suspects. We agree with the importance of this definition and we acknowledge that we failed to communicate effectively the ways in which our new procedure addresses this criterion. The revised manuscript corrects this problem, we believe, and we detail here how we came to this conclusion, since it is based on types of data and analyses that are fundamentally different from those that are associated with traditional lineup experiments. We hope that the reviewers will carefully consider the following arguments.

To claim improvement as defined by the reviewers we need to know two things for a given decision criterion: the probability of correctly identifying the target and the probability of incorrectly identifying an innocent suspect. Using traditional lineups these two probability measures are obtained independently using two separate experiments. In a target-present experiment the target is included in the lineup and the choices made by a population of subjects reveal the probability of correctly identifying the target. In a target-absent lineup the target is not included in the lineup, but one (or any) lineup face is a “designated suspect.” Similar to the target-present lineup, choices made by a population of subjects in the target-absent lineup reveal the probability of incorrectly identifying an innocent suspect.

The logic underlying this methodological approach to traditional lineups is well founded and has stood the test of time. But even as our goals are the same, we believe this is not the best approach to data we have collected using the method of paired comparisons. Despite these differences, we present (here and in the revised manuscript) a sequence of steps that highlight analogies to the analysis methods employed for traditional lineups. By this means we illustrate how we obtained the two key probability measures (correct ID of target and incorrect ID of innocent) for every decision criterion. We begin by highlighting a couple of important points about the design of our experiment and the structure of our data.

There are two important methodological innovations incorporated in our approach, which have not heretofore been applied to the problem of eyewitness identification. The first of these is the ability to scale recognition memory signals elicited by each lineup face. This scaling process

estimates recognition memory signals and yields a set of six probability distributions. Each distribution identifies the probability that one of the lineup faces is the “cause” of a given recognition memory signal. The second methodological innovation is the use of a statistical decision procedure known as ideal observer analysis. The “ideal observer” is a statistical classification device, which we use to determine the probability of correct and incorrect IDs for every possible decision criterion, based on the scaled measures of recognition memory. We note that there is nothing particularly unusual about either the scaling procedure that we have used or the procedure for statistical classification. The novelty and benefits of these tools lies in their unprecedented joint application to the problem of eyewitness identification.

We used the ideal observer approach to measure the probability of correctly identifying the target, i.e., to classify faces as target or non-target, based on the scaled probability distributions for each face produced by the method of paired comparisons. We performed a similar analysis to obtain a measure of the probability of incorrectly identifying an innocent suspect, with one procedural exception: In this case the target was removed from the analysis.

This target-removal approach naturally raises an important question: how do we know that perceptual scaling of non-targets is the same regardless of whether the target is present or absent? The answer to this question comes from a branch of statistics and decision theory that looks at robustness of scaling procedures (often applied, for example, in efforts to optimize voting rules¹). Specifically, the target-removal procedure is legitimate if it satisfies the stringent mathematical criterion of “local independence of irrelevant alternatives” (LIIA). This criterion is met when perceptual scaling is unaffected by removal of the top vote-getter (the target in our case). Critically, paired comparison scaling procedures are among the few methods that satisfy this LIIA requirement.² In other words, the outcome of scaling the five non-targets is unaffected by the absence of the target. It follows that data obtained from those five non-targets is sufficient to evaluate, using the ideal observer approach, the probability that an innocent suspect will be incorrectly identified.

Thus, by the means described above, we have in hand the two key probability measures (for correct and incorrect IDs) and we have those measures for all possible decision criteria (which is another advantage of our procedure). Perhaps most importantly, we have the complete set of measures for every individual subject, which is yet another advantage of our procedure, highlighted extensively in the manuscript. We combine these probability measures (separately for the population and for individual subjects) in the form of ROC curves in Figs 6 and 7 of the revised manuscript. The ordinate in each ROC plot is the probability of correct target ID and the abscissa is the probability of incorrect ID of an innocent suspect. We have termed this form of presentation a “Recognition ROC” curve because it derives (per method outlined above) from the scaled recognition memory signals. The area under this curve (conventionally abbreviated as AUC) is a criterion-free index of the degree to which the ideal observer is capable of correctly distinguishing between the target and an innocent suspect, based on scaling data.

We stress here that the Recognition ROC conveys the same characteristics of performance that are commonly reported in studies of traditional SIM and SEQ lineups, but does so by exploiting the full structure of recognition memory for lineup faces while avoiding the ambiguity of witness’

¹ See for example, H.P. Young (1995). Optimal Voting Rules, *Journal of Economic Perspectives*, 9, 51-64; D. Luce and H. Raiffa (1957). *Games and Decisions*; K. Arrow and E. Maskin (1951). *Social Choice and Individual Values*; J. Nash (1950). The Bargaining Problem, *Econometrica*. 18, 155–162; P. Tannenbaum (2010). Mathematics of Voting, *Excursions in Modern Mathematics*, 7th Ed.

² H. P. Young (1988). Condorcet's Theory of Voting, *American Political Science Review* 82, 1231-124; H. P. Young and A. Levenglick (1978). A Consistent Extension of Condorcet's Election Principle, *SIAM Journal on Applied Mathematics* 35, 285–300.

decision criteria for identification. Furthermore, using a bootstrap procedure (see Methods of the revised manuscript) we have estimated variance associated with each Recognition ROC, which positions us to make statistical evaluations of performance relative to other lineup types. Because witness performance also depends upon lineup fairness, however, we must make assumptions about the fairness of our lineup relative to comparable simultaneous or sequential lineups.

One additional advantage of our PAR method is that perceptual scaling yields a quantitative index of lineup fairness (roughly proportional to slope of the voting score function). From our scaling data (Fig 4 of revised manuscript), we deem our lineup to be unfair in that there are significant differences in perceptual similarity of each lineup face to the target. We thus chose to compare PAR performance with traditional lineup performance from a study in which the lineup was acknowledged to be unfair (Experiment 2 of Mickes et al., 2012). This comparison, shown in new Fig 4, reveals that for a range of decision criteria PAR and traditional lineups yield comparable performance, with an important caveat that the relative degrees of lineup fairness are unknown in these studies (as is generally true of inter-study comparisons of lineup performance). We acknowledge that this is but a preliminary comparison based on data from different laboratories, but we note that the fairness measure yielded by perceptual scaling opens a new door to systematic quantitative studies of the effects of lineup fairness on probability of correct vs incorrect identification.

2) The paired comparison lineup procedure is not novel.

REVIEWER 2:

...the authors are apparently unaware that this approach of using a continuous measure is definitely not new. Whether one uses similarity ratings or confidence, there are well-designed experiments that examine eyewitness performance that do not ask the witness to make a yes/no type decision. The work of Neil Brewer and his lab is a dominant example. And, the work of Fitzgerald & Price (2016) and more recently Bruer and Price (2017) uses a so-called "face off" procedure that involves pairwise comparisons. So, there is a literature out there that is highly similar to this so-called "novel" procedure and any eyewitness scientist who knows the literature knows that this is not so novel.

AUTHORS' RESPONSE:

The work that we have described is not simply a "continuous measure" approach designed to overcome the limitations of "yes/no type decisions." As we note in the revised manuscript, our approach employs a powerful combination of perceptual scaling and signal detection analysis, which has numerous advantages (see revised manuscript and Theme 1, above) and has not heretofore been applied to the problem of eyewitness identification. In the remainder of our response to this concern, we detail the ways in which our approach differs from those of the specific investigators cited by Rev 2,^{3,4} and why these differences are important.

Comparison with Price and Fitzgerald "face-off" procedure: Although there is a partial procedural similarity between the lineup presentation method employed in the "face-off" paradigm described by Price and colleagues⁵ and the method that we have described, there are several important differences between the goals, the underlying theory, the data analyses, and the utility of the findings. To appreciate these differences, it's important to first recall that our goal in the present experiment was to estimate the strengths of recognition memory signals for

³ Brewer, N., Weber, N., Guerin, N. (2019). Police lineups of the future? *American Psychologist*. Advance online publication.

⁴ Price HL, Fitzgerald RJ (2016) Face-off: A new identification procedure for child eyewitnesses. *Journal of Experimental Psychology. Applied*. **22**(3), 366-380.

⁵ Ibid.

each lineup face so that it would be possible for an ideal observer to make a criterion-independent judgment about the significance of any lineup ID. The Thurstonian paired comparison procedure is a natural choice for this perceptual scaling approach to recognition memory, and has (as we note in the revised manuscript) been employed to address many analogous problems of perceptual scaling in the real world. The beauty of this PAR approach as applied to eyewitness identification is that it takes the ultimate decision out of the hands of the eyewitness. The decision is made by a process of statistical inference, in which an “ideal observer” classifies faces as target or non-target based on the estimated memory signals, and thereby overcomes the fundamental ambiguity of the witness’ decision criterion.

By contrast, the “face-off” procedure as practiced involves only a limited number of face pairings. Price and Fitzgerald employ a procedure in which 8 face stimuli are used.⁶ There are $(8 \times 7)/2 = 28$ possible face pairings from this set, but only 7 of these unique pairs are presented to the witness using this method, and each pair is presented only once. The face-off procedure is a very limited version of a voting scheme that was developed initially by the 18th century French mathematician Nicolas de Condorcet.⁷ The complete form of Condorcet’s voting scheme – in which all possible pairs are presented – has highly advantageous ranking properties⁸ and it is a foundation of Thurstone’s psychophysical scaling method and, by extension, the PAR procedure we have employed.⁹ Most importantly, we note that the face-off procedure, as applied by Price and Fitzgerald, is NOT capable of yielding an estimate of either the strength or the variance of recognition memory signals for each of the lineup faces and thus cannot be used to estimate the significance of any identification. Nor was any of this the purpose for which the face-off procedure was intended by Fitzgerald and Price.

One is nonetheless compelled to ask how much stronger the identification evidence is for the winner in the face-off task, relative to other faces in the lineup. Witnesses may express a confidence level for their judgment in this face-off task, but such criterion-dependent measures are among the problems that our approach seeks to overcome. We also note that the face-off procedure assumes transitivity of Condorcet voting without evidence that this assumption holds. To put it simply, suppose that face A is voted more similar to the target than face B, and likewise face C more than D. If we then pair the two semifinalist winners, A and C, and find that A wins, there is little ground to assume that subjects would rank A more similar to the target than D – a pairing that has never been tested.¹⁰ This lack of transitivity is known as the Condorcet paradox. Our Thurstonian paired comparison approach avoids the assumption of transitivity by presentation of all possible pairs. The face-off procedure does not.

Comparison with “continuous measures” approach of Brewer et al.: We are indeed familiar with this work, reported most recently in a 2019 paper in *American Psychologist*.¹¹ The approach involves obtaining a confidence measure for each lineup face that reflects the subject’s decision about similarity of each face to the target. The commendable purpose of this

⁶ Ibid.

⁷ in his *Essay on the Application of Analysis to the Probability of Majority Decisions*, 1785.

⁸ H. P. Young (1995). Optimal Voting Rules, *Journal of Economic Perspectives*, 9, 51-64.

⁹ The limited Condorcet approach characteristic of the Price and Fitzgerald face-off procedure may be more familiar to some as an “elimination game bracket,” whereas the complete Condorcet method characteristic of the PAR lineup is analogous to a “round-robin game bracket.”

¹⁰ To illustrate this notion in real world terms, consider the “final four” of this year’s NCAA “March Madness,” Virginia beat Auburn and Texas Tech beat Michigan State. The two semifinal winners, Virginia and Texas Tech, then played in the national championship game, which Virginia won to take the title. Virginia was never paired with 2nd ranked Michigan State and there is certainly no guarantee that Virginia would have won this game, even though transitivity would predict so.

¹¹ Brewer, N., Weber, N., Guerin, N. (2019). Police lineups of the future? *American Psychologist*. Advance online publication.

approach was to gain insights into recognition memory signals for all lineup faces (i.e. those below the surface of a thresholded “he’s the one” response characteristic of traditional lineups). Although our purpose is similar, the Brewer approach is not a precedent for the method we have proposed.

To fully appreciate this, it is important to recognize that the Brewer method and the Thurstonian paired comparison method map, respectively, onto two established approaches for evaluating differences between sensory stimuli: Affective and Discriminative. Affective methods (e.g. preference and confidence ratings) are inherently noisy because they rely upon criterion-dependent judgments. Discriminative methods are preferred because they offer greater precision of measurement – they directly assess the ability to discriminate between (usually pairs of) stimuli in a forced-choice task, a process well known to manifest little criterion dependence.¹² The method of Brewer is Affective in that it relies upon subjective confidence ratings, which is a weakness that our Discriminative perceptual scaling/signal detection approach seeks to avoid.

By contrast, our approach is based upon forced-choice judgments of similarity to a remembered target, from which we estimate both mean and variance of recognition memory signals. As we have emphasized in our manuscript and elsewhere in our responses to reviewers (see Theme 1, above), the beauty of our PAR approach is that the resulting voting score probability distributions constitute evidence that can be employed for classification of faces as target vs. non-target. Unlike the Brewer approach, our analysis also provides a measure of the statistical significance of the top-ranked face, and does so for individual subjects, in the form of the area under the ROC curve.

In sum, while the Affective confidence-based approach of Brewer and others provides greater information than traditional lineup approaches, it does not employ the most important features of our Discriminative approach, namely perceptual scaling from a forced-choice paradigm to reveal the full set and moments of recognition memory signals, followed by ideal observer analysis of discriminability.

3) The new paired comparison lineup procedure is advertised as overcoming the confound between response bias and memory strength, but recent ROC methods do the same.

REVIEWER 1:

The motivation for designing the PAR lineup arose from concerns about response bias and memory strength. The authors wrote, “Traditional lineup procedures rely on overt eyewitness responses that confound two covert factors: the strength of recognition memory and the criterion for deciding what perceptual evidence is sufficient for reporting a match.” This is an issue when the diagnosticity ratio is used, however, ROC analysis measures discriminability independently from response bias (e.g., Rotello & Chen, 2016; Wixted & Mickes, 2012). The authors should consider acknowledging this.

AUTHORS’ RESPONSE:

This previous use of ROC analysis is acknowledged in our revised manuscript. It is important to note, however, that the analysis employed in previous reports using ROC depends upon “expressed confidence level” as a proxy for response bias. Confidence judgments of this sort

¹² See Lesmes LA, Lu Z-L, Baek, J, Tran N, Doshier BA, Albright (2015). Developing Bayesian adaptive methods for estimating sensitivity thresholds (d') in Yes-No and forced-choice tasks, *Journal of Vision*. Also see Luce (1963) and Green & Swets (1966).

are notoriously criterion dependent and can vary widely from subject to subject.¹³ Our method avoids this problem by estimating the recognition memory signals to all of the lineup faces and then applying an ideal observer analysis to reveal the discriminability of target vs. non-target faces as a function of decision criterion. This is a powerful approach rooted in a well-established perceptual scaling/signal detection model and to the best of our knowledge no previous study has adopted this perceptual scaling approach to the eyewitness problem.

It's also worth emphasizing here, as we have done in the revised manuscript and above under Theme 1, that previous applications of ROC analysis to the eyewitness problem use data collected from traditional simultaneous and sequential lineups, in which each subject contributes but one data point. The analysis is thus necessarily based on a large population of subjects. Because our perceptual scaling/signal detection analysis can be based on estimates of the recognition memory signals for each subject (witness) to all of the lineup faces employed, we have been able to apply ROC analysis to data collected from individual subjects. As we note in our manuscript, the area under each subject's ROC curve can be employed as a quantitative measure of ID certainty and thus an index of the probative value of the eyewitness evidence.

4) The characterization of decision models of recognition memory is confusing.

REVIEWER 3:

I found the characterization of decision models of recognition confusing. The authors continually refer to the 'implicit response' and 'implicit decision criterion' of subjects. I think the former is referring to the evidence upon which decisions are based. If so, 'implicit response' is confusing for two reasons. First, recognition memory evidence, or at least one component of recognition memory evidence, is assumed explicit, not implicit (viz., contextual recollection). Indeed, I believe the authors use the term 'declarative memory' in the text which, of course, assumes subjects can report the contents of memory in an overt fashion. The second aspect of 'implicit response' that is confusing, if the authors are referring to recognition evidence, is that 'response' implies a decision that has been rendered.

However, under statistical decision models the evidence doesn't dictate the form or nature of the decision, which instead reflects a decision operation that is often modeled to optimize, in some statistical fashion, the translation of evidence into judgment. There are similar problems with the notion of an 'implicit decision criterion.' Decision criteria or thresholds are usually assumed to be at least partially under subject control, which is why subjects can adopt a more or less conservative approach when instructed or given payout matrices. For both of these phrases, the authors may be mixing the idea that the decision and evidence process are unknown/covert/, with the idea they are implicit. The latter traditionally is taken to mean unconscious and involuntary and I don't think this is the authors' intent but I'm not certain.

AUTHORS' RESPONSE:

The reviewer is indeed correct – we employed badly chosen terms to convey our points about memory and decision criteria. We did not mean to suggest that they are unconscious or involuntary, and we are embarrassed to realize that the term “implicit” conveyed that meaning. What we intended to refer to is what Thurstone (1927) called the “discriminal process” that characterizes the observer's underlying reaction to a sensory event. Thurstone was explicitly agnostic about the nature of this process, since that nature was viewed as irrelevant to his Law of Comparative Judgment, but he noted that it may take the form of neuronal signals. Consistent with this view and with Rev 3's point, we have removed “implicit” from the relevant parts of the manuscript, and we now use “recognition memory signals” to refer to the discriminational

¹³ Roediger, H. L. III, Wixted, J. H., & DeSoto, K. A. (2012). The curious complexity between confidence and accuracy in reports from memory. In L. Nadel & W. P. Sinnott-Armstrong (Eds.), *Oxford series in neuroscience, law and philosophy. Memory and law* (pp. 84-118). New York, NY, US: Oxford University Press.

processes elicited by our face stimuli, and “decision criterion” to refer to a threshold for deciding whether a given discriminial process is the result of one stimulus or another.

5) The method for estimating variance of recognition memory signals to target and non-target faces (as employed for the individual subject analysis) needs clearer and more concrete explanation.

REVIEWER 3:

While I find the method of paired comparison interesting, it might be helpful if the authors could explain a little more concretely how the performance can be taken to illustrate the variance of targets and lures.

AUTHORS' RESPONSE:

We have clarified this procedure in the methods section of our report where we have separately explained how we handled data in the target-present and target-absent analyses of the PAR data.

6) Does the PAR method require the assumption of equal voting variance for targets and non-targets? If so, is that assumption compatible with the ROC analysis?

REVIEWER 3:

If I understand the methods section correctly, it is necessary to assume that the target and lures have equal variance in order to use the method to place the stimuli along a single dimension. If so, then this is a potential problem because the single item, confidence based recognition ROC is inconsistent with the equal variance assumption, leading to the unequal variance signal detection model (e.g., Mickes, Wixted & Wais 2007), and/or dual process models (e.g., Yonelinas 1994; Mickes & Wixted 2010) that do not make this assumption. Of course, if the assumption of equal variance isn't valid, then the scaling solution the method of paired comparison produces in this domain is spurious.

...As noted above, the PAR method seems to require that one assume target and lure evidence distributions are equally variable; an assumption that seems unwarranted given the recognition ROC literature. This means that one of the potential strengths of the procedure, the ability to provide bias free subject accuracy measures (Figure 4B) is undercut to the degree this assumption is critical. Is there some way the authors can test how critical this assumption is? For example, does the rank ordering of the individuals, in terms of AUC, change appreciably when the target distribution is assumed up to 1.5 times more variable than the lures? I imagine that if a simulation demonstrated that the rank ordering of AUCs across individuals was remarkably stable when target variability was randomized within a reasonable range (given the prior literature) it would be quite newsworthy. Conversely, if the rank ordering of AUCs heavily depends on the equal variance assumption (or the equivalence of the target variance across participants) then value of the AUC estimation would be undercut.

... it is unclear how critical the equal variance assumption is to the modeling of individual subject ROCs... at a minimum it seems important to demonstrate that the individual ROCs are largely impervious to unequal variance possibilities...

...I imagine that if they could demonstrate ... that the rank ordering of individual accuracy estimates is not particularly dependent upon the equal variance assumption...then they would have a pretty influential study.

AUTHORS' RESPONSE:

REV 3 is indeed correct that the method used to estimate voting variance for individual subjects (see Theme 5) requires the assumption that variance is equal for all faces. This is of course the standard assumption for ROC analysis. As REV 3 notes, single-item confidence-based recognition ROC is inconsistent with the equal variance assumption and previous investigators have developed unequal variance models to address this. Because our method does not rely upon single-item confidence-based recognition ROC – we are simply using ideal observer

analysis to classify stimuli based on equal variance probability distributions – this is not a concern.

7) The instructions given to lineup participants are not adequately explained.

REVIEWER 3:

The instructions given to the participants are not covered in sufficient detail. Currently, it is unclear what they were led to believe about the presence of the perpetrator in the lineup procedure. This is critical since in both the sequential and simultaneous procedures subjects can reject the lineup. In contrast, the PAR method they cannot.

AUTHORS' RESPONSE:

In all three lineup types, subjects were told that the suspect may or may not be in the lineup. We have added text to the methods in order to make this clear. A lineup rejection is certainly possible with the PAR procedure, as noted in the revised manuscript methods, and as noted by REV 3 in a later paragraph (see Theme 8 below). This happens when two or more faces elicit the same recognition memory signals, which we presume to be one of the reasons that a subject would reject a simultaneous or sequential lineup (though unlike the PAR procedure, those reasons are not transparent). In our experiment, PAR lineup rejections occurred for 5/62 (8%) subjects (see Fig 3A), which is markedly less frequent than for simultaneous and sequential lineups (see Fig 3B and C). We speculate below (see Theme 8) about the utility of this knowledge and the causal underpinnings.

8) Lineup rejections for simultaneous and sequential procedures are qualitatively different than those for the PAR procedure.

REVIEWER 3:

Perhaps more problematic is that the authors test the number of rejections of the lineup across the three procedures. However, these data are non-comparable because rejections in the PAR procedure have nothing to do with observers deciding that they do not want to risk identifying an innocent suspect...they must select a member of the pair on each trial. Instead, it appears that 'rejections' here reflect subjects for which two or more probes have the top rank in terms of selections. These different rates strike me as simply different things, and they are clearly calculated fundamentally differently. Given this, the statistical test strikes me as uninformative.

AUTHORS' RESPONSE:

A lineup rejection in the PAR procedure happens when two or more faces elicit the same (see below for definition of “sameness”) recognition memory signals. We assume that this is a frequent underlying cause for a lineup rejection in a simultaneous or sequential lineup, although we note that neither of these traditional lineup types afford the level of transparency given by the PAR data. To phrase this point about PAR lineup rejection in the reviewer’s own terms: Yes, “rejections in the PAR procedure have nothing to do with [subjects] deciding that they do not want to risk identifying an innocent suspect,” but such rejections have everything to do with the ideal observer not “wanting” to identify an innocent suspect. In short, it is the ideal observer (not the witness) that provides the basis for lineup rejection using the PAR procedure.

The difference in frequency of lineup rejections across lineup types is striking and we feel that a statistical test is appropriate, since we believe it to be based on the same underlying cause and it conveys something important about the relative utility of the different lineup types.

The more interesting question here is why the PAR lineup elicits fewer rejections. We note broadly that the frequency of lineup rejections is directly tied to the criterion for deciding identity of recognition memory signals to two (or more) lineup faces. This identity criterion is largely

uncontrolled and unknown in the case of simultaneous and sequential lineups, and presumably varies significantly across subjects (though it is surely influenced by the strength and variance of memory signals, witness instructions and biases). Using our combination of perceptual scaling and signal detection approaches, any given identity criterion can be mapped to a specific area under the Recognition ROC curve. Observed AUC values below that value would be cause for lineup rejection. We have applied a stringent identity criterion to define a rejection: the recognition memory signals must be numerically identical (within the limits of our measurement ability), such that the AUC=0.5. We could of course loosen this identity criterion, but the effect of doing so is revealed by the ROC analysis itself.

9) What is the effect of prior probability on lineup performance?

REVIEWER 3:

Given that the discrimination performance of the simultaneous and sequential group depends upon beliefs about the prior probability the perpetrator is present (ideally), it would be useful to compare the methods across a range of biases regarding the likelihood the perpetrator is present in the lineup. Regardless, providing the exact instructions given to all three groups seems important.

AUTHORS' RESPONSE:

The question here is about base rate assumptions made by lineup participants. As noted above (see theme 7), we have edited our methods to clarify that all lineup participants in each condition were given the same instructions about the presence or absence of a suspect in the lineup. The reviewer suggests that it would be useful to compare lineup performance across different base rate assumptions. We agree and we may pursue this in future studies, but we feel that this is beyond the scope of the present report.

10) Statistical tests of the PAR outcome are unnecessary.

REVIEWER 3:

The authors note that the suspect was the top ranked choice in the PAR procedure, but this would necessarily be true unless memory was non-existent and it doesn't seem to require a statistical test.

AUTHORS' RESPONSE:

Contrary to the reviewer's assumption, the suspect (target) being the top choice in the PAR procedure is not a logical necessity. What we sought to test using our perceptual scaling/signal detection approach is (1) what is the "top-ranked" face, and (2) how discriminable is the estimated response to the top-ranked face from responses to the other five lineup faces. With regard to the first point, 32% of our PAR subjects had the target as the top-ranked face, meaning that the majority of subjects in fact had top-ranked faces that were not the target. We used a simple statistical test to show that the frequency of top-ranked targets (32%) is greater than would be expected by chance (17%), and we feel that this test is appropriate under the circumstances. However, as we note in the manuscript, looking only at top-ranked status is a superficial treatment of an extremely rich data set. Which is – with regard to the second point about discriminability – the reason why we performed the ROC analysis on population data for recognition memory signals to target and non-target faces (see Figs 6 and 7, and Theme 1, above).

11) The authors explicitly ask whether the PAR method provides useful information about the identity of the culprit. This seems a foregone conclusion.

REVIEWER 3:

The first question [posed by the authors] is whether the PAR method provides useful information about the identity of the culprit. However, I fail to see how it cannot unless the observers had no memory of the culprit. Analogously, it would be odd to ask if the sequential procedure provided useful information about the identity of the culprit when it was introduced, since again, how could it not if observers have some memory for the culprit.

AUTHORS' RESPONSE:

REV 3 is correct, of course, and our only defense is that the question was rhetorical. The real question (also rhetorical in some sense) is whether the Thurstonian technique translates well to the eyewitness context. At any rate, our revised text no longer poses this question.

12) In what sense does the PAR result allow inferences about utility of eyewitness testimony?

REVIEWER 3:

The second question was whether the PAR results allow one to draw inferences about the utility of that witnesses testimony. I wasn't entirely certain how this question differed from the first, but I believe the authors are highlighting that the procedure may give a bias free estimate of accuracy for each individual. I think this is a notable goal and would be an advance over methods that allow subjects to reject the lineup as a whole, but presently it is not clear if this is what the authors are asking here.

AUTHORS' RESPONSE:

The reviewer's inference is correct: the question we posed in this case is about utility of the PAR method for individual subjects (witnesses). The answer, as noted in the manuscript, is that the method has great utility in this respect and that this is one of the important ways in which the PAR lineup is distinguished from the traditional simultaneous and sequential lineups. To see why this is true, recall that application of ROC analysis to simultaneous and sequential data necessarily requires that each subject contribute but one piece of choice data (since each subject is presented with only one lineup and makes only one decision). Thus the traditional lineup ROC necessarily requires a population analysis. This population-based approach has demonstrated value for comparing the average performance of subjects in simultaneous vs. sequential paradigms, thus helping to make decisions about which type of lineup to employ. But traditional lineups cannot be used to evaluate performance of any individual witness. By contrast, "Recognition ROC" curves resulting from our PAR lineup analysis can be derived independently from each subject's data, which is possible because Thurstone's scaling procedure yields individual-subject estimates of both means and variances of recognition memory signals to each lineup face. As we note in the manuscript, this advance is both unprecedented and powerful: The resulting area under the Recognition ROC curve is a quantitative measure of certainty for a given witness, which can be used as an index of probative value of that witness' testimony. We have clarified these points about procedure and value in the revised manuscript.

13) Why is the target always ranked highest when averaged across subjects?

REVIEWER 3:

Interestingly, the authors demonstrate that across subjects who rank the filler highest, the target nonetheless has the highest mean rank. I believe this simply means that the lineup is fair among the fillers such that errors do not arise because one is preferentially similar to the suspect (i.e., a patsy).

AUTHORS' RESPONSE:

This is an interesting question. We were initially surprised by this outcome when the analysis was restricted to subjects who individually ranked a non-target highest (Fig 3F). Literally, this means that the target was ranked highly by all subjects and with little within-subject voting

variance, even in cases in which it was NOT the top-ranked face. This finding is important to our argument since it is one of the insights gained by employing an approach that allows us to look beneath the surface of a traditional lineup ID.

We believe that REV 3 is correct that a fair lineup is a sufficient cause for this outcome, because a high-rank and highly consistent vote for a non-target face would be evidence of an unfair lineup and would likely have led to a non-target averaging highest rank. At the same time, we note that our method offers quantitative insight into the definition of a fair lineup (see Theme 1, above), since perceptual similarity is exactly what the Thurstonian scaling method measures. We have added a note to this effect in our description of the population analysis results.

REVIEWER 3:

While this [the counterintuitive finding that the individual subjects can be incorrect but their average is correct] is an interesting characteristic, it would presumably also arise for the other procedures if the observers were forced to rank the lineup members. Namely, that when averaged across those who ranked a lure as highest, the sample mean of the target would nonetheless be higher (i.e., an item analysis would show the highest mean for the target). Finally, the second-choice procedure in forced-choice procedures seems germane. Under this approach, during say four alternative forced choice recognition, observers are required to make a second choice after their first selection, corresponding to their guess if their first response were incorrect. Here, second choice accuracy is higher than chance, which presumably means that when averaged across subjects, the target in this sub-sample nonetheless has the highest item mean (e.g., Parks & Yonelinas 2009).

AUTHORS' RESPONSE:

We believe that the reviewer is correct on all of these points, and in that sense we acknowledge that our seemingly counterintuitive observation about individual vs average rankings is not all that surprising. We mentioned this observation in our manuscript simply because we felt that it warranted some explanation, which we provided. More to the point of our report, however, we stress that none of the procedures cited in this reviewer comment has the advantages that the Thurstonian paired comparison method demonstrably has when applied to the problem of eyewitness identification (i.e. criterion-independent decision making based on estimates of mean and variance of recognition memory signals).

14) Is it really necessary to use ROC analysis to demonstrate that subjects select the target more often than non-targets?

REVIEWER 3:

The authors perform a statistical comparison across a population ROC representing discrimination of the target face from all others (F1 vs ~F1) versus an ROC with classes of the second most selected face (F3) and all others (except F1 and F3). I'm not sure what this test answers that is not already clear simply by directly looking at the ranks or selection rates the authors already show in Figure 3D. In other words, this seems like a pretty complex way to ask whether or not the subjects select the target more often than the next most highly selected item, but perhaps I am missing why ROCs are uniquely suited to answer this question.

AUTHORS' RESPONSE:

We believe that what the reviewer may be asking here is: Why not just use a couple of simple t-tests to show that the difference in memory strength between the top-ranked face and the other faces is greater when the top-ranked face is the target vs. a non-target? This is certainly doable, but it would neglect all of the insights that signal detection analysis affords generally, such as clear intuitions about classification of face stimuli (and thus their discriminability), across the spectrum of decision criteria. The ROC method also allows us to compute AUC, which we offer as a quantitative index of certainty. Finally, the use of ROC analysis places our approach in line with recent methods for evaluating eyewitness performance using traditional

simultaneous and sequential lineups. At any rate, the ROC analysis, as we have employed it, is neither technically nor conceptually complex.

15) The known relationship between accuracy and confidence is not correctly stated in the Discussion.

REVIEWER 3 (“MINOR”):

In the Discussion the authors discuss confidence during simultaneous and sequential procedures noting confidence is collected ‘based on the assumption that a confident witness is more likely to make an accurate identification.’ However, this is not really an assumption, but an established fact. As a whole, more confident judgments are more accurate than less confident judgments in this literature and as Wells, Wixted, and Roediger have noted (e.g., Wixted & Wells 2017), accuracy conditioned on rendering the highest possible confidence, immediately after selection, is quite high...and invariably higher than accuracy when confidence is lower.

AUTHORS’ RESPONSE:

The reviewer is correct and we are remiss in not clearly stating that under specific circumstances there is an empirically established correlation between confidence and accuracy in the eyewitness literature. We have corrected this in the text. Nonetheless, the point of our comment was to emphasize that the area under the ROC curve (AUC) for individual subjects is potentially of greater value as an objective quantitative index of certainty than is the subject’s expressed confidence level. One of the reasons why this is true is that the established relationship between confidence and accuracy for traditional lineups is based on data averaged across many subjects. For any given subject that relationship may vary, largely because confidence judgments are heavily criterion-dependent. All one can ever do is make a prediction about accuracy based on the average. The AUC metric, by contrast, leads to a prediction based on data from the individual subject in question.

16) There is a problem with the way ROC data are presented in Fig 4.

REVIEWER 3 (“MINOR”):

One of the individual subject ROCs doesn’t appear to use 15 criteria (black) although the authors indicate that 15 thresholds were used.

AUTHORS’ RESPONSE:

The reviewer is correct. We failed to explain our procedure in the original manuscript, and we apologize about the confusion. Since distributions of voting scores are discrete and they vary between subjects, not every decision criterion produces a distinct point in the individual ROCs. In the revised manuscript we prevented this confusion by removing the criterion markers from the plots of ROCs (now Fig 7).

REVIEWER 3 (“MINOR”):

Additionally, it wasn’t quite clear how the 15 [criteria] were determined.

AUTHORS’ RESPONSE:

We used equally spaced decision criteria that span the full range of target and non-target frequency distributions (i.e. from the minimum non-target value to the maximum target value). Given the discrete distribution of voting scores (with the maximum vote per face being five and three repetitions of each face pair) 15 values of decision criteria provided sufficient resolution to clearly reveal the shape of the ROC curve.

17) Why are different numbers of subjects employed for different lineup types?

REVIEWER 3 (“MINOR”):

Is there some reason the authors use twice the number of subjects for the PAR as the simultaneous and sequential procedures?

AUTHORS’ RESPONSE:

The primary purpose of the present report, as noted in the manuscript and detailed above in response to questions about the comparison of PAR and traditional lineups (see Theme 1), was to convey the results of a practical evaluation of a promising new lineup approach that has several principled advantages over traditional lineups. In fact, the only data and analyses that we present for simultaneous and sequential lineups appear in Fig 3 and the corresponding part of the main text. In this case, we only compare raw frequencies of correct target selection and lineup rejection, as is common in eyewitness lineup studies, using a simple statistical test. Because our focus was on the new PAR procedure, we felt that a smaller number of subjects for the simultaneous and sequential lineups was sufficient to make these points. This is supported by the fact that randomly subsampling the PAR data in Fig 3, to yield a subject number more similar to that for simultaneous and sequential data, produces the same result when subjected to the same simple frequency analyses.

18) Why not collect confidence ratings from subjects in the PAR lineup?**REVIEWER 3 (“MINOR”):**

Why not use confidence during the PAR? I don’t see any reason why this would contaminate the procedure and it might allow a method of comparison to the other procedures.

AUTHORS’ RESPONSE:

We believe that the reviewer is correct in assuming that there is little or no cost associated with collection of expressed confidence ratings from subjects who participate in the PAR lineup condition. Indeed, we have collected these data and a preliminary analysis reveals only a weak relationship to the AUC measure of certainty. We suspect that this is due to the fact that subjects offered a confidence judgment after presentation of each face pair, which collectively (i.e. on average) may not be a good measure of overall confidence. At any rate, we are exploring this further but we feel that it is beyond the scope of the present report.

Reviewers' comments:

Reviewer #1 (Remarks to the Author):

I appreciate the authors' responses to the reviewers' comments, the clarifications, and the revisions made to the manuscript. In response to one of my comments, the authors wrote, "It is important to note, however, that the analysis employed in previous reports using ROC depends upon 'expressed confidence level' as a proxy for response bias." While most reports have used confidence levels to plot the ROCs it is not necessary. For example, Mickes et al. (2017) compared confidence-based ROCs with ROCs constructed by plotting correct ID rate and false ID rate pairs from different biasing-instructions. Of course, that does not get around the concern that the authors have with criteria.

In the response to reviewers' letter, the authors wrote, "Using traditional lineups these two probability measures are obtained independently using two separate experiments." However, this would be bad practice. Participants are typically randomly assigned to a target-present or target-absent lineup in the same experiment. I do not think the authors meant it as it sounds, but I wanted to set the record straight just in case.

I recommend publication. The major appeal of this new lineup procedure is the fact that it is based on long-standing scientific techniques. It should be in the literature for researchers to assess and interrogate its strengths and weaknesses and debate whether the new lineup would indeed be a viable replacement of traditional lineup procedures. Will it stand up to the scrutiny? Maybe these techniques will be more useful for perception vs. memory, but it's worth a try.

Reviewer #3 (Remarks to the Author):

This is the second review of Gepshtein and colleagues 'A perceptual scaling approach to eyewitness identification.' In my initial review I focused on potential drawbacks of the methods and I found confusing regarding their interpretation and claims. In their cover letter the authors have been quite responsive and detailed. The manuscript is also clearer although I suspect some of the confusion I experienced in the original will remain for readers who have not been privy to the review and hence cannot, or would not have time, to consult the responses in the cover letter to the revision. Overall, I think the authors make a strong case that the Thurstonian scaling method offers promise for eyewitness recognition research, and that it garners considerably more information than is possible with single report procedures. That said, I continue to feel that the claims that the procedure is manifestly superior to extant approaches still strike me as overstated for the reasons I outline below.

In my initial review I and others noted that the procedure had not been established as superior to the considered alternatives because it had not been tested in the target absent situation. The authors' response was that a target absent condition is unnecessary because the recalculation of the voting scores with the target removed satisfies a LIAA criterion, in which the scaling (presumably among the

lures) doesn't change when the top vote getter is removed. This however confuses a statistical with a psychological question. The fact that the calculated scaling solution for the lures doesn't change when the target is removed, in no way demands that the subjects' approach to the task cannot change under actual testing conditions in which no target was present. Currently, we simply have no way of knowing if the scaling solution is completely identical for a new batch of subjects tested without a target and hence the critique that the current data doesn't show superiority in both instances seems to remain. Thus while I agree the procedure appears to offer many benefits in the target present case, it is not clear that it does so in the target absent case.

The authors stress that one advantage of this approach is that it uses ideal observer analysis. However, I still am confused as to what they specifically mean by this. If they are referring to the use of ROCs to summarize the accuracy of the observers, then I concur that the ability to do so is a nice aspect of the procedure. However, 'ideal observer' usually is taken to mean an optimal decision rule in some regard and so the claim is confusing. Instead, the current procedure is able to provide a criterion free estimate of accuracy, which again, is great...but doesn't strike me as an ideal observer decision model in the traditional sense of the Neyman-Pearson statistical decision framework underlying SDT. Under this approach one determines an ideal decision criterion as a function of a cost-benefit assignment to various outcomes. Normally, I wouldn't focus that much on terminology but the authors continue to assert that the current procedure is superior to extant methods because of the scaling procedure and the use of *ideal observer* methods. Perhaps it would suffice to simply note early on that the Thurstonian scaling procedure affords the use of ROCs which are by construction, criterion free measures of discriminability.

In the prior review I noted that the Thurstonian scaling procedure the authors use seems to require that one assume that the target and lures have equal evidence variability. The authors responded that this is correct, but because their method doesn't use confidence based ROCs, and instead uses an ideal observer, this is not a concern. I'm afraid however that I don't understand why not. The fit of the unequal variance confidence-based ROC has been taken to suggest that the underlying memory evidence is itself more variable for targets than lures. This conclusion may be wrong but the adoption of a Thurstonian scaling procedure that *requires* that one assume equal variance in evidence in lures and targets doesn't demonstrate that it is wrong. I assume that the PAR procedure assumes equal variance because there simply isn't enough information in the data to provide a separate variance estimate for each stimulus along the hypothetical scale, or to test whether doing so reliably improves the solution. Thus my prior question was directed at how serious this simplifying assumption is for the resulting accuracy estimate if there is greater evidence variance for targets, and this question remains unaddressed in the current revision.

In summary, I think the authors have made a strong case that Thurstonian scaling should be explored in the context of eyewitness performance and they clearly demonstrate that it is capable of yielding a criterion free estimate of accuracy for each observer. That said, the current version still strongly implies that it is superior to extant methods without empirical data directly demonstrating this across target present and target absent conditions and showing that detections and lineup rejections are improved in

both instances. Moreover, it continues to minimize the potentially negative effects of assuming that the evidence for targets and lures is equally variable in strength. Thus, while I am enthusiastic about the potential utility of the method and think these data demonstrate that potential nicely, I nonetheless think the paper still occasionally veers into asserting that which remains to be empirically tested and demonstrated.

Response to Reviewer Concerns

Once again, we are grateful for the time and expertise that the reviewers have brought to our manuscript. We are heartened by numerous reviewer statements of support for our research strategy and our discoveries:

R1: I appreciate the authors' responses to the reviewers' comments, the clarifications, and the revisions made to the manuscript.

R1: I recommend publication. The major appeal of this new lineup procedure is the fact that it is based on long-standing scientific techniques. It should be in the literature for researchers to assess and interrogate its strengths and weaknesses and debate whether the new lineup would indeed be a viable replacement of traditional lineup procedures.

R3: In their cover letter the authors have been quite responsive and detailed. The manuscript is also clearer.

R3: Overall, I think the authors make a strong case that the Thurstonian scaling method offers promise for eyewitness recognition research, and that it garners considerably more information than is possible with single report procedures.

R3: I think the authors have made a strong case that Thurstonian scaling should be explored in the context of eyewitness performance and they clearly demonstrate that it is capable of yielding a criterion free estimate of accuracy for each observer.

R3: I am enthusiastic about the potential utility of the method and think these data demonstrate that potential nicely.

We nonetheless appreciate the remaining concerns. We address these concerns below by grouping them by content. In each case we first briefly state what we understood to be the problem. This is followed by the corresponding reviewer text and concluded in each case by our response.

Reviewer Concern #1: Our statement about methods for determining decision criterion in experiments using traditional simultaneous and sequential lineups was incomplete.

R1: In response to one of my comments, the authors wrote, "It is important to note, however, that the analysis employed in previous reports using ROC depends upon 'expressed confidence level' as a proxy for response bias." While most reports have used confidence levels to plot the ROCs it is not necessary. For example, Mickes et al. (2017) compared confidence-based ROCs with ROCs constructed by plotting correct ID rate and false ID rate pairs from different biasing-instructions. Of course, that does not get around the concern that the authors have with criteria.

Author response: The reviewer is correct. Decision criterion has often been estimated from expressed confidence, as we noted. But as the reviewer suggests, it can also be (and has been) manipulated directly using instructions. We have clarified this point in the revised manuscript.

Reviewer Concern #2: We inappropriately referred to different experiments when we actually meant to refer to different conditions of the same experiment.

R1: In the response to reviewers' letter, the authors wrote, "Using traditional lineups these two probability measures are obtained independently using two separate experiments." However, this would be bad practice. Participants are typically randomly assigned to a target-present or target-absent lineup in the same experiment. I do not think the authors meant it as it sounds, but I wanted to set the record straight just in case.

Author Response: The reviewer is correct. We meant to say that the two probability measures are traditionally obtained independently using two separate conditions of the same experiment, not two separate experiments. We have corrected this in the revised manuscript.

Reviewer Concern #3: A target-absent lineup condition is necessary to accurately measure the probability of misidentifying an innocent suspect. Therefore, without data from a target-absent condition we cannot make a convincing case that the PAR lineup compares well to traditional SIM and SEQ lineup procedures.

R3: In my initial review I and others noted that the procedure had not been established as superior to the considered alternatives because it had not been tested in the target absent situation. The authors' response was that a target absent condition is unnecessary because the recalculation of the voting scores with the target removed satisfies a LIAA criterion, in which the scaling (presumably among the lures) doesn't change when the top vote getter is removed. This however confuses a statistical with a psychological question. The fact that the calculated scaling solution for the lures doesn't change when the target is removed, in no way demands that the subjects' approach to the task cannot change under actual testing conditions in which no target was present. Currently, we simply have no way of knowing if the scaling solution is completely identical for a new batch of subjects tested without a target and hence the critique that the current data doesn't show superiority in both instances seems to remain. Thus while I agree the procedure appears to offer many benefits in the target present case, it is not clear that it does so in the target absent case.

R3: the current version still strongly implies that it is superior to extant methods without empirical data directly demonstrating this across target present and target absent conditions and showing that detections and lineup rejections are improved in both instances.

ED: Reviewer 1 echoes Reviewer 3's concerns [re target absent condition].

R2 *FROM PRIOR REVIEW*: The bottom line is that this experiment would never be accepted by eyewitness identification scientists as a test of which lineup procedure is superior because it fails to include the target-absent conditions.

Author Response: Per reviewer recommendations, we have extended our experiment to include a target-absent (TA) condition. As detailed in the revised manuscript, the TA condition differed from the target-present (TP) condition in that the target face was removed from the lineup and replaced by a face that matched – like all of the other non-target faces – the physical description of the target. We ran 70 new subjects in this TA condition. The data obtained reveal, not surprisingly, that perceptual scaling of the non-target faces in the TA lineup is, on average, similar to scaling of non-target faces in the TP lineup (cf. Figs 4A and 6 of revised manuscript). (As we note in the revised manuscript, the correlation between non-target voting scores in TP and TA conditions was 0.86; $p=0.036$.)

As recommended by reviewers, we used the new TA PAR lineup data, in conjunction with our existing TP PAR lineup data, to compute a population ROC curve that reveals the average probabilities of correct vs. incorrect identifications as a function of decision criterion. This new Recognition ROC appears in Fig 7 of the revised manuscript. There are three points that we wish to make here about this new analysis and plot:

(1) As we emphasize in the revised manuscript, our new Recognition ROC stands up well to comparisons with published data from experiments using traditional lineups, further supporting our assertion that the PAR approach has great potential as a new tool for eyewitness identification.

(2) The new Recognition ROC is nearly identical to the Recognition ROC that we presented in our previous version of the manuscript, which was derived from an analysis of the “simulated” target-absent condition (the AUC values are 0.75 for the new ROC and 0.70 for the previous ROC). Our previous approach was based on the argument that data for the five non-target faces in the TP lineup could be analyzed independently of the target to obtain a measure of the probability of identifying an innocent suspect. This argument is theoretically sound and has many merits, as we

noted previously. The observed similarity between results obtained using the “actual” and “simulated” TA conditions adds critical support to this argument. We nonetheless recognize the importance – as stressed by the reviewers – of empirically validating our new procedure using the tried and true TA methods that have been employed for traditional lineups. We have done so, as reflected in the revised manuscript.

(3) One of the strengths of the PAR approach is that it yields perceptual scaling data for individual witnesses (subjects), which cannot possibly be obtained using traditional SIM and SEQ lineups. In principle, individual subject data can be used in conjunction with signal detection analysis – analogous to our population data approach – to reveal a criterion-independent measure of the ability of a given witness to discriminate target vs. innocent suspect, which would be of enormous value to the trier of fact. The problem with this plan is that individual subject data can only be obtained for one PAR lineup condition – a single subject cannot be included in both TP and TA conditions. In other words, we cannot usefully employ an *actual* TA lineup to estimate the probability that a given subject will identify an innocent suspect.

A solution to this problem is afforded by the simulated TA condition that we employed in the previous version of our manuscript (summarized under #2, above). Our new use of an “actual” TA lineup validates, at the population level, both the perceptual scaling and binary classification outcomes of the simulated TA approach. (See additional arguments in the revised manuscript.) For this reason, we have chosen to retain the simulated TA condition for the analysis of single-subject data, in order to take advantage of the unprecedented opportunity to gain insight into identification accuracy by individual witnesses. The revised manuscript summarizes the arguments to support this novel approach.

Reviewer Concern #4: Our use of the term “ideal observer analysis” is confusing.

R3: The authors stress that one advantage of this approach is that it uses ideal observer analysis. However, I still am confused as to what they specifically mean by this.

Author Response: “Ideal observer analysis” is a term of art, which we now recognize to have different meanings to different research communities. Our use of the term follows from Geisler (*Vision Research*, 2011): “An ideal observer is a hypothetical device that performs a given task at the optimal level possible, given the available information and any specified constraints.” The task in our case is simply one of binary classification of decisions based on two overlapping probability distributions of behavioral responses. As used in our previous manuscript, the “ideal observer” was meant to refer to the best possible criterion-independent binary classification algorithm. Given the apparent ambiguity of the term “ideal observer,” and given that the method we have employed is a standard feature of signal detection analysis, we have replaced the term “ideal observer analysis” with the term “signal detection analysis.” The principles of the method – an optimal statistical classification algorithm – remain explicit in the revised manuscript.

Reviewer Concern #5: Are our assumptions about variance of underlying probability distributions valid and how do they affect conclusions about identification performance.

R3: In the prior review I noted that the Thurstonian scaling procedure the authors use seems to require that one assume that the target and lures have equal evidence variability. The authors responded that this is correct, but because their method doesn’t use confidence based ROCs, and instead uses an ideal observer, this is not a concern. I’m afraid however that I don’t understand why not. The fit of the unequal variance confidence-based ROC has been taken to suggest that the underlying memory evidence is itself more variable for targets than lures. This conclusion may be wrong but the adoption of a Thurstonian scaling procedure that *requires* that one

assume equal variance in evidence in lures and targets doesn't demonstrate that it is wrong. I assume that the PAR procedure assumes equal variance because there simply isn't enough information in the data to provide a separate variance estimate for each stimulus along the hypothetical scale, or to test whether doing so reliably improves the solution. Thus my prior question was directed at how serious this simplifying assumption is for the resulting accuracy estimate if there is greater evidence variance for targets, and this question remains unaddressed in the current revision.

Author Response: We apologize for our previous misguided response to this question. The reviewer raises what we now understand to be an interesting and reasonable concern. The underlying motivation for this concern is the fact that many decades of studies of list memory (e.g. Egan 1958; Ratcliff et al. 1992; Wixted 2007) have routinely obtained evidence for a model in which variance of the target memory distribution is greater than variance of the lure memory distribution – i.e. the unequal-variance model of recognition memory. It turns out, however, that the unequal-variance model is empirically inconsistent with eyewitness data generally. Target and filler memory distributions in eyewitness studies – in cases where this has been examined – are best described by equal-variance models. See Wixted et al. (*Cognitive Psychology*, 2018, **105**, 81-114) for a comprehensive treatment of this issue, along with speculations about why studies of list memory and eyewitness memory yield different results.

We have assumed equal variance for target and filler distributions for precisely the reasons that the reviewer has identified. As noted in the previous paragraph, this assumption is entirely consistent with prior empirical studies of eyewitness memory, i.e., there does not appear to be “greater evidence variance for targets” than for fillers.

***REVIEWERS' COMMENTS:

Reviewer #1 (Remarks to the Author):

I was pleased to see that the authors decided to heed the recommendations of the reviewers and test memory using their new procedure on target-absent lineups. I am not sure what “relatively high degree of performance” means in this sentence, “Nonetheless, the relatively high degree of performance evinced by the PAR lineup in these inter-study comparisons that our novel approach has significant potential as a tool for eyewitness identification.” It may come across as meaning that the PAR is better, which isn't the point of that plot. But I like the rest of the sentence, “novel approach has significant potential as a tool for eyewitness identification”. I agree. And that is why I recommended publication before and still do (it's even better now with the inclusion of the target-absent data).

Reviewer #3 (Remarks to the Author):

This is the third review of Gepshtein and colleagues ‘A perceptual scaling approach to eyewitness identification.’ I remain highly enthusiastic about the report and strongly believe it will make an important contribution to the literature and spur a considerable amount of new research. It is a novel and exciting approach. Additionally, the authors now have provided an additional data set examining preferences for the lures in the absent of any target. These data seem to roughly support the idea that the relative preferences of lures is independent of the presence or absence of the target, but I don't think they strongly support that contention as discussed below. I do not think this is a serious problem but I would recommend that the authors add a sentence or two acknowledging that the preferences are not perfectly preserved, but that they appear to hold for the most confusable lures. Regardless, I do not need to review the manuscript again and I hope it is publicly available soon.

Addressing a prior concern, the authors have now included a target absent data set to examine the relative selection probabilities of the lures used in the target present data. This was done to ensure that the simulation method of determining individual ROCs, that is uniquely possible with the PAR procedure, can legitimately use the voting scores of the lures of the lures in a target present condition as a proxy for a target absent situation. It is clear from their Figure 6 of these new data that the rank order of the stimuli is roughly the same as in the target present group, and they show this similarity by calculating a correlation of .86 on (I believe) the five pairs of scores. However, Figure 6 suggests that Lure 2 was reliably preferred over Lure 4, yet the original data in Figure 4a indicates the opposite. Doesn't this demonstrate that the relative preferences of the lures is not constant across target absent and target present conditions? I do not think this invalidates the PAR or their approach to estimating individual ROCs (which is great), but to me it suggests that it is important to note that the LIA assumption only approximately holds. Presumably despite this small violation (if it is one) the rank ordering of the subjects in terms of their individual ROC areas is still diagnostic of relative accuracy across subjects.

MINOR CONCERNS

The approach to simulating the individual ROCs is fairly clear. However, one small question remains. I believe the simulation allows locating each probe on a continuum and estimating the variance around each of these locations assuming a normal distribution. Does this mean that the ROCs plotted for each individual are between the target distribution and the closest lure on the continuum? In other words, I am not sure how the N-1 different estimates of the lure distributions are being contrasted with the target distribution by sweeping a criterion from most to least stringent voting score.

This is a *very* minor concern. The authors use the term 'decision criterion' when discussing simultaneous and sequential line ups, which reflects how much evidence each individual requires to make a selection as opposed to rejecting the lineup. They also rightfully note that the PAR procedure itself is criterion free. However, when discussing the plotting of population and individual ROCs they discuss the performance of the group or each individual in terms of a decision criterion. I realize that these are not the same constructs and that the former is a psychological criterion and the latter a methodological one. However, readers might be confused by the use of a methodological criterion to plot the ROC in a task that doesn't require the subject to use a psychological criterion. I'm not sure what language would best avoid potential confusion but perhaps a small footnote highlighting the distinction might help.

Response to Reviewer Concerns

Once again, we are grateful for the time and expertise that the reviewers have brought to our manuscript. We are heartened by reviewer statements of support for our research. In the following we address the remaining comments and identify the manner in which we have accordingly revised the manuscript.

Reviewer Comment #1: We used ambiguous terms to describe eyewitness performance.

R1: I am not sure what “relatively high degree of performance” means in this sentence, “Nonetheless, the relatively high degree of performance evinced by the PAR lineup in these inter-study comparisons that our novel approach has significant potential as a tool for eyewitness identification.” It may come across as meaning that the PAR is better, which isn't the point of that plot. But I like the rest of the sentence, “novel approach has significant potential as a tool for eyewitness identification”. I agree. And that is why I recommended publication before and still do (it's even better now with the inclusion of the target-absent data).

Author Response: We thank the reviewer for drawing our attention to the loose language with which we characterized eyewitness performance evidence by the ROC plot in Fig 7. We have modified the offending sentence accordingly in the revised manuscript.

Reviewer Comment #2: The addition of a target-absent condition is beneficial. The report is novel and exciting and an important contribution to the literature.

R1: I was pleased to see that the authors decided to heed the recommendations of the reviewers and test memory using their new procedure on target-absent lineups.

R2: I remain highly enthusiastic about the report and strongly believe it will make an important contribution to the literature and spur a considerable amount of new research. It is a novel and exciting approach. Additionally, the authors now have provided an additional data set examining preferences for the lures in the absent of any target.

Author Response: We thank the reviewers for this support and encouragement. We agree that the inclusion of the additional data set on target-absent lineups makes for a stronger presentation.

Reviewer Comment #3: Observed perceptual scaling of lineup fillers is not perfectly identical in target-present and target-absent conditions. Variations and corollaries of this point are raised in several related comments listed and addressed independently below.

R2: These data seem to roughly support the idea that the relative preferences of lures is independent of the presence or absence of the target, but I don't think they strongly support that contention as discussed below. I do not think this is a serious problem but I would recommend that the authors add a sentence or two acknowledging that the preferences are not perfectly preserved, but that they appear to hold for the most confusable lures.

Author Response: The reviewer is correct on two counts:

- (1) The reviewer correctly observed that the scaling of lineup fillers is slightly different between target-present (TP) and target-absent (TA) lineups. As noted in the manuscript

there are five fillers that are in common for both TP and TA lineups. We expected that the perceptual scaling of these would be identical regardless of the presence or absence of the target. This is largely true, in that the order of fillers in the TP condition is 3, 6, 5, 4, 2, whereas it is 3, 6, 5, 2, 4 in the TA condition. The only ordinal difference is the flip between fillers 4 and 2. This flip is significant in the sense that the 4 vs. 2 difference is significant in both cases. We have added a note to this effect in the revised manuscript.

- (2) The reviewer correctly notes that the scaling order between TP vs. TA lineups “holds for the most confusable lures.” This is a critical point because it means that the 4 vs. 2 reversal has negligible effect on the performance of our binary classifier. Indeed, as we have noted, the AUC values for actual TA and simulated TA analyses are 0.77 and 0.70, respectively. We have added a statement to the revised manuscript to make the reviewer’s point about the most confusable lures.

R2: Addressing a prior concern, the authors have now included a target absent data set to examine the relative selection probabilities of the lures used in the target present data. This was done to ensure that the simulation method of determining individual ROCs, that is uniquely possible with the PAR procedure, can legitimately use the voting scores of the lures of the lures in a target present condition as a proxy for a target absent situation. It is clear from their Figure 6 of these new data that the rank order of the stimuli is roughly the same as in the target present group, and they show this similarity by calculating a correlation of .86 on (I believe) the five pairs of scores.

Author Response: This is essentially the same point raised in the preceding comment, and we note again that the reviewer is correct. In addition to the aforementioned revisions to address this, we have also added clarification that the correlation of 0.86 is indeed the correlation between the five pairs of voting scores for the five fillers that are included in both TP and TA lineups.

R2: However, Figure 6 suggests that Lure 2 was reliably preferred over Lure 4, yet the original data in Figure 4a indicates the opposite. Doesn’t this demonstrate that the relative preferences of the lures is not constant across target absent and target present conditions?

Author Response: This is again the same point raised above about the reversal of scaling order for fillers 4 and 2 between TP and TA lineups. The reviewer is correct and we have addressed as indicated above.

R2: I do not think this invalidates the PAR or their approach to estimating individual ROCs (which is great), but to me it suggests that it is important to note that the LIIA assumption only approximately holds. Presumably despite this small violation (if it is one) the rank ordering of the subjects in terms of their individual ROC areas is still diagnostic of relative accuracy across subjects.

Author Response: This is related to the point raised above about the reversal of scaling order for fillers 4 and 2 between TP and TA lineups. The reviewer is correct that this is a “small violation” of the order predicted by Limited Independence of Irrelevant Alternatives (LIIA), as we note in the revised manuscript. The reviewer is also correct that such small violations, to the extent that they occur, have negligible effect on the individual ROCs.

Reviewer Comment #4: A clarification is needed about derivation of individual ROCs.

R2: MINOR CONCERN – The approach to simulating the individual ROCs is fairly clear. However, one small question remains. I believe the simulation allows locating each probe on a continuum and estimating the variance around each of these locations assuming a normal distribution. Does this mean that the ROCs plotted for each individual are between the target distribution and the closest lure on the continuum? In other words, I am not sure how the N-1 different estimates of the lure distributions are being contrasted with the target distribution by sweeping a criterion from most to least stringent voting score.

Author Response: We sought to make the method used to derive individual ROCs as similar as possible to the method we used to derive population ROCs. Thus, the Correct Identification Rate (plotted on the ordinate in each panel of Fig 8) was derived from the target-present condition for the indicated subject. The False Identification Rate (plotted on the abscissa) was derived from the simulated target-absent analysis for the indicated subject (obtained as we explained above) using all five lure distributions (rather than the second-ranked lure alone). We have clarified this issue in Methods (section “Individual ROC analysis of eyewitness data”). We note that this is also explained in the first paragraph of the Results section “Population Analysis of Eyewitness Data” and in the fifth paragraph of the Results section “Individual Subject Analysis of Eyewitness Data.”

Reviewer Comment #5: The term “decision criterion” is used in both a psychological and a methodological sense, which may cause reader confusion.

R2: This is a *very* minor concern. The authors use the term ‘decision criterion’ when discussing simultaneous and sequential line ups, which reflects how much evidence each individual requires to make a selection as opposed to rejecting the line ups, which reflects how much evidence each individual requires to make a selection as opposed to rejecting the lineup. They also rightfully note that the PAR procedure itself is criterion free. However, when discussing the plotting of population and individual ROCs they discuss the performance of the group or each individual in terms of a decision criterion. I realize that these are not the same constructs and that the former is a psychological criterion and the latter a methodological one. However, readers might be confused by the use of a methodological criterion to plot the ROC in a task that doesn’t require the subject to use a psychological criterion. I’m not sure what language would best avoid potential confusion but perhaps a small footnote highlighting the distinction might help.

Author Response: The reviewer is correct, though we note that this potential ambiguity of the term “decision criterion” would apply to any application of signal detection analysis to human decisions informed by vision and memory. We have attempted to clarify our use of the term in the revised manuscript.